# Plasmodium vivax chloroquine resistance links to *pvcrt* transcription in a genetic cross

Juliana M. Sá[1], Sarah R. Kaslow [1], Roberto R. Moraes Barros[1], Nicholas F. Brazeau[2], Christian M. Parobek[3], Dingyin Tao [4], Rebecca E. Salzman[1], Tyler J. Gibson [1], Soundarapandian Velmurugan[5], Michael A. Krause[1], Viviana Melendez-Muniz[1], Whitney A. Kite[1], Paul K. Han[1], Richard T. Eastman [1], Adam Kim[6], Evan G. Kessler[1], Yonas Abebe[5], Eric R. James[5], Sumana Chakravarty[5], Sachy Orr-Gonzalez[7], Lynn E. Lambert[7], Theresa Engels[8], Marvin L. Thomas[8], Pius S. Fasinu[9], David Serre [6], Robert W. Gwadz[1], Larry Walker [9], Derrick K. DeConti[10], Jianbing Mu[1], Jeffrey A. Bailey [9,11], B. Kim Lee Sim[5], Stephen L. Hoffman[5], Michael P. Fay [12], Rhoel R. Dinglasan[4,13], Jonathan J. Juliano[2,3,14] & Thomas E. Wellems [1]

Mainstay treatment for *Plasmodium vivax* malaria has long relied on chloroquine (CQ) against blood-stage parasites plus primaquine against dormant liver-stage forms (hypnozoites), however drug resistance confronts this regimen and threatens malaria control programs. Understanding the basis of *P. vivax* chloroquine resistance (CQR) will inform drug discovery and malaria control. Here we investigate the genetics of *P. vivax* CQR by a cross of parasites differing in drug response. Gametocytogenesis, mosquito infection, and progeny production are performed with mixed parasite populations in nonhuman primates, as methods for *P. vivax* cloning and in vitro cultivation remain unavailable. Linkage mapping of progeny surviving >15 mg/kg CQ identifies a 76 kb region in chromosome 1 including *pvcrt*, an ortholog of the *Plasmodium falciparum* CQR transporter gene. Transcriptional analysis supports upregulated *pvcrt* expression as a mechanism of CQR.

[1] Laboratory of Malaria and Vector Research, National Institute of Allergy and Infectious Diseases, National Institutes of Health, Bethesda, MD 20892, USA. [2] Department of Epidemiology, Gillings School of Global Public Health, University of North Carolina, Chapel Hill, NC 27599, USA. [3] Curriculum in Genetics and Molecular Biology, School of Medicine, University of North Carolina at Chapel Hill, Chapel Hill, NC 27599, USA. [4] W Harry Feinstone Department of Molecular Microbiology and Immunology, Johns Hopkins Bloomberg School of Public Health, Baltimore, MD 21205, USA. [5] Sanaria Inc., Rockville, MD 20850, USA. [6] Institute for Genome Sciences, University of Maryland School of Medicine, Baltimore, MD 21201, USA. [7] Laboratory of Malaria Immunology and Vaccinology, National Institute of Allergy and Infectious Diseases, National Institutes of Health, Bethesda, MD 20892, USA. [8] Division of Veterinary Resources, Office of Research Services, National Institutes of Health, Bethesda, MD 20892, USA. [9] Department of Pharmaceutical Sciences, College of Pharmacy and Health Sciences, Campbell University, Buies Creek, NC 27506, USA. [10] Program in Bioinformatics and Integrative Biology, University of Massachusetts Medical School, Worcester, MA 01655, USA. [11] Division of Transfusion Medicine, Department of Medicine, University of Massachusetts Medical School, Worcester, MA 01655, USA. [12] Biostatistics Research Branch, National Institute of Allergy and Infectious Diseases, National Institutes of Health, Rockville, MD 20852, USA. [13] Emerging Pathogens Institute, Department of Infectious Diseases & Immunology, College of Veterinary Medicine, University of Florida, Gainesville, FL 32611, USA. [14] Division of Infectious Diseases, School of Medicine, University of North Carolina at Chapel Hill, Chapel Hill, NC 27599, USA. Correspondence and requests for materials should be addressed to T.E.W. (email: twellems@niaid.nih.gov)

Malaria cases from *Plasmodium vivax* infection number an estimated 14.3 million annually and are often life threatening[1]. Mainstay treatment for more than half a century has been chloroquine (CQ) for blood-stage parasitemia followed by primaquine to eradicate persistent liver-stage (hypnozoite) infection[2]. Spreading drug resistance now confronts this regimen and threatens malaria control programs, particularly in Papua New Guinea and Indonesia where artemisinin-based combination therapies are instead recommended[3]. Molecular understanding of resistance will provide information in support of malaria control and antimalarial drug discovery.

Features of CQR differ between *P. vivax* and *Plasmodium falciparum*, the main agent of malaria morbidity and mortality in Africa. CQ-resistant (CQ-R) *P. falciparum* strains spread through Southeast Asia and South America in the 1960s, entering Africa in the late 1970s and causing catastrophic resurgences of morbidity and mortality[4], whereas *P. vivax* CQR was not reported until the late 1980s in Papua New Guinea[5]. *P. falciparum* CQR results from mutations in the *pfcrt* (*P. falciparum* CQ resistance transporter) gene, including a change in codon 76 that protects the parasite against CQ accumulation and damage[6,7]. In contrast, *P. vivax* CQR could not be attributed to codon mutations in *pvcrt* (also termed *pvcrt-o*, *pvcg10*), the *P. vivax* ortholog of *pfcrt*, neither in a study of CQ treatment failures in humans and monkeys[8], nor in recent genome wide association surveys[9,10]. These findings left open the possibility that *pvcrt* expression levels have a role in resistance.

Heterologous expression experiments have shown that *pvcrt* introduced by transfection can affect CQ response in *P. falciparum*[11] and alter CQ accumulation in *Dictyostelium discoideum*[11] and yeast[12]. Increased transcript levels of *pvcrt* and the *P. vivax* multidrug resistance gene (*pvmdr-1*) have been reported in samples from patients with severe *P. vivax* malaria[13] or who suffered CQ treatment failures[14]. In contrast, another study found that *pvcrt* transcription was related to parasite stage but not to ex vivo CQ susceptibility in patient samples[15]. The lack of practical methods to routinely cultivate, isolate clones, and genetically manipulate parasites in the laboratory continues to limit direct studies of the CQR mechanism in *P. vivax*. Experimental investigations have consequently relied on nonhuman primates (NHP), such as *Aotus* or *Saimiri* New World monkeys, and a limited number of *P. vivax* lines adapted to these models. Here we report the production and analysis of a genetic cross of *P. vivax* parasites having different CQ response phenotypes. Our findings identify upregulated expression of the *pvcrt* gene as a mechanism of drug resistance.

## Results

**Design of a *P. vivax* laboratory genetic cross.** Our experiments initially explored the use of sporozoites from infected *Anopheles* mosquitoes to inoculate *Aotus nancymaae* or *Saimiri boliviensis* monkeys, but multiple attempts failed to produce adequate progeny infections. These attempts included ten different monkey-adapted *P. vivax* lines (NIH-1993, Vietnam-IV, AMRU-I, Indonesia-XIX, Indonesia-I/CDC, Brazil-I/CDC, Salvador-II, North Korea, Panama, and Chesson) by blood feedings to ca. 15,000 mosquitoes of available species: *Anopheles dirus* (South East Asia), *Anopheles albimanus* (Central America), *Anopheles stephensi* (India), *Anopheles freeborni* (North America), and *Anopheles gambiae* (Africa). Some parasite lines did not infect the mosquitoes (through direct or membrane feeding) despite observation of healthy-appearing gametocytes; other lines infected mosquitoes and yielded sporozoites but failed to produce blood-stage infections after inoculation of the sporozoites into the monkeys. We therefore turned to a splenectomized chimpanzee

as host for the cross. This strategy depended upon two separate inoculations of the individual animal: (i) a joint inoculation of parental *P. vivax* blood-stage parasites to produce infective male and female gametocytes for cross-fertilization in *Anopheles* mosquitoes; and (ii) subsequent inoculation of cryopreserved sporozoites, to infect the liver with a variety of progeny that subsequently enter the chimpanzee bloodstream (Fig. 1a).

**CQ selection of distinct gametocyte subpopulations.** Two *P. vivax* lines were chosen as candidate parents for inoculation of the chimpanzee: CQ-R AMRU-I that survives a total of 30 mg/kg CQ in *Aotus*[16], and NIH-1993, closely related to, but genetically distinct from, the Salvador-I line (Supplementary Table 1), which typically does not survive this treatment dose in *Aotus*[17] (Supplementary Fig. 1, Supplementary Table 2; recommended CQ treatment for adult humans is 25 mg/kg total dose delivered orally over 3 days[2]). NIH-1993 and AMRU-I-parasitized erythrocytes were obtained from individual infections of *Aotus* monkeys, sterilely washed, resuspended, and inoculated into the chimpanzee as a mixture of $5 \times 10^6$ blood-stage parasites from each line. Thirteen days post blood inoculation (DPBI), asexual parasites and gametocytes were observed in thin blood smears (Supplementary Fig. 2a, Supplementary Table 3), but polymerase chain reaction (PCR) assays 18 DPBI detected only microsatellite alleles of the NIH-1993 line (Supplementary Fig. 2b). Therefore, 21 DPBI, the chimpanzee was given a sub-therapeutic single dose of CQ (2.7 mg/kg) to increase the representation of any CQ-R parasites. This uncovered no AMRU-I parasites, indicating that the AMRU-I line had failed to adapt and multiply in the chimpanzee; however, the CQ dose increased the representation of an NIH-1993 subpopulation distinguishable from the major CQ-sensitive (CQ-S) subpopulation by a variety of microsatellite markers (Supplementary Table 4). These subpopulations, identified as NIH-1993-S and NIH-1993-R, were present in comparable numbers 28 and 31 DPBI and therefore served as parents of the genetic cross.

**Generation of NIH-1993 S × R recombinant sporozoites.** *A. stephensi*, *A. freeborni*, *A. gambiae*, and *A. dirus* mosquitoes were infected on six occasions with the mixed NIH-1993-S and NIH-1993-R gametocytes, both by direct feeding on the chimpanzee and by laboratory feedings of blood samples through membranes, to produce recombinant sporozoites (Supplementary Fig. 2, Supplementary Table 5; Feeds 1–6). Single oocysts from mosquito midguts presenting both alleles of distinguishing parental markers suggested cross-fertilization of NIH-1993-S and NIH-1993-R gametes from feeds after the 2.7 mg/kg CQ dose (Supplementary Fig. 2c; Feeds 3–6). Further, either of two types of circumsporozoite protein (CSP) VK210 or VK247 were found in the sporozoites from mosquito salivary glands (Fig. 1b, c), indicating inheritance of distinct alleles from the parents. Samples of aseptically isolated sporozoites were cryopreserved and confirmed to be infective in hepatocyte invasion assays[18] (Supplementary Table 5). The chimpanzee was cured 34 DPBI with Malarone (confirmed by follow-up PCR).

**Development of NIH-1993 S × R recombinant blood stage progeny.** Six months after cure, the chimpanzee was intravenously inoculated with the NIH-1993-S × NIH-1993-R sporozoites from Feeds 4 and 5. NIH-1993 S × R blood-stage progeny from this cross were microscopically detected 8 days post sporozoite inoculation (DPSI; Supplementary Table 6). Blood was collected 15 and 18 DPSI (Progeny Pools 1 and 2), and the chimpanzee was treated with CQ 5 mg/kg/day × 3 days. Bloodstream parasitemia declined to levels detected only by PCR, then

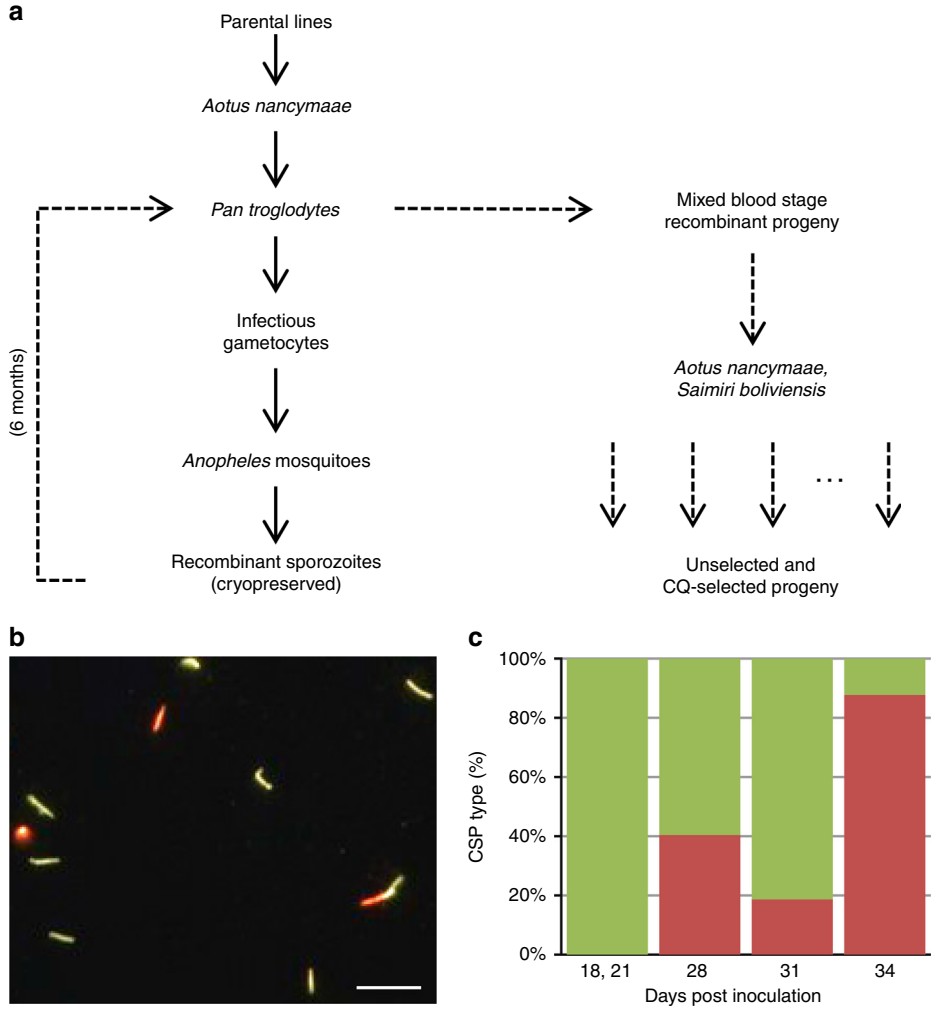

**Fig. 1** Steps in the production of a *P. vivax* genetic cross and generation of recombinant progeny. **a** Work flow of the cross. Blood-stage parasites of *P. vivax* parental lines were collected from *Aotus* monkeys and co-inoculated into a *Pan troglodytes* chimpanzee. After development of parasitemia, chimpanzee blood samples containing infectious gametocytes were fed on multiple occasions to *Anopheles* mosquitoes to allow cross-fertilization of parental lines. Recombinant sporozoites from mating events were recovered from the mosquitoes and cryopreserved, and the chimpanzee was treated to eliminate any remaining *P. vivax* parental parasites. The recombinant sporozoites were inoculated into the chimpanzee 6 months after cure of parental lines infection to recover mixed blood-stage progeny from the cross. Pools of recombinant progeny were collected and sub-inoculated into *Aotus* and *Saimiri* monkeys for phenotypic characterization and genetic analysis. Solid arrows indicate the first chimpanzee infection inoculated with blood-stages from the parental lines, and dotted arrows the second infection with recombinant progeny sporozoites. **b** Immunofluorescence assays were performed on recombinant sporozoites with antibodies specific for the VK210 and VK247 versions of *P. vivax* CSP expressed by distinct parental lines. Sporozoites expressing *P. vivax* CSP VK210 type are represented in green and sporozoites expressing VK247 are represented in orange. Scale bar indicates ~25 µm. **c** Percentages of *P. vivax* sporozoites expressing CSP VK210 and VK247 in the infected mosquitoes. Four batches of cryopreserved sporozoites were obtained from *An. stephensi*, *An. freeborni*, *An. dirus*, and *An. gambiae* mosquitoes fed 18, 21, 28, 31, and 34 days after the co-inoculation (sporozoites from 18 and 21 feedings were combined and presented in the first bar). The chimpanzee was treated with a low dose of CQ (2.7 mg/kg) on day 21 to increase CQ-R *P. vivax* parental population carrying the VK247 marker and improve chances of cross-fertilization over self-mating events in the mosquito

increased to 0.01% in thin blood films 43 DPSI. After an additional 3-day CQ treatment of 2.5 mg/kg/day beginning 53 DPSI, parasites were detectable by PCR until final cure at 102 DPSI with Malarone plus primaquine; follow-up PCR testing confirmed no remaining infection after one year.

**NIH-1993 S × R progeny differ in CQ response.** Samples of the unselected mixed progeny were inoculated into 20 *Aotus* or *Saimiri* monkeys for drug response analysis (Supplementary Fig. 3). Parasitemia developed in 17 of these monkeys. After CQ treatment (15–55 mg/kg total dose), 10/17 of these infections recrudesced, in many cases after the parasitemias had decreased below microscopically detectable levels (Supplementary Fig. 4,

Supplementary Table 7). HPLC-mass spectroscopy of plasma samples from these monkeys confirmed therapeutic CQ levels ($\geq$10 ng/mL)[2], with a single exception (Supplementary Table 8).

**Linkage-group selection maps a chromosome 1 determinant.** Infected blood from each monkey before CQ administration (NIH-1993 S × R unselected progeny) and after recrudescence (selected progeny) provided 10 sample pairs for linkage group selection analysis (LGS)[19]. Quantitative inheritance signals were obtained from these LGS pairs by PCR with 37 polymorphic microsatellite markers distributed irregularly across the 14 *P. vivax* chromosomes (Supplementary Fig. 5). CQ selection *E* values displayed by heat map (Fig. 2a, Supplementary Table 9),

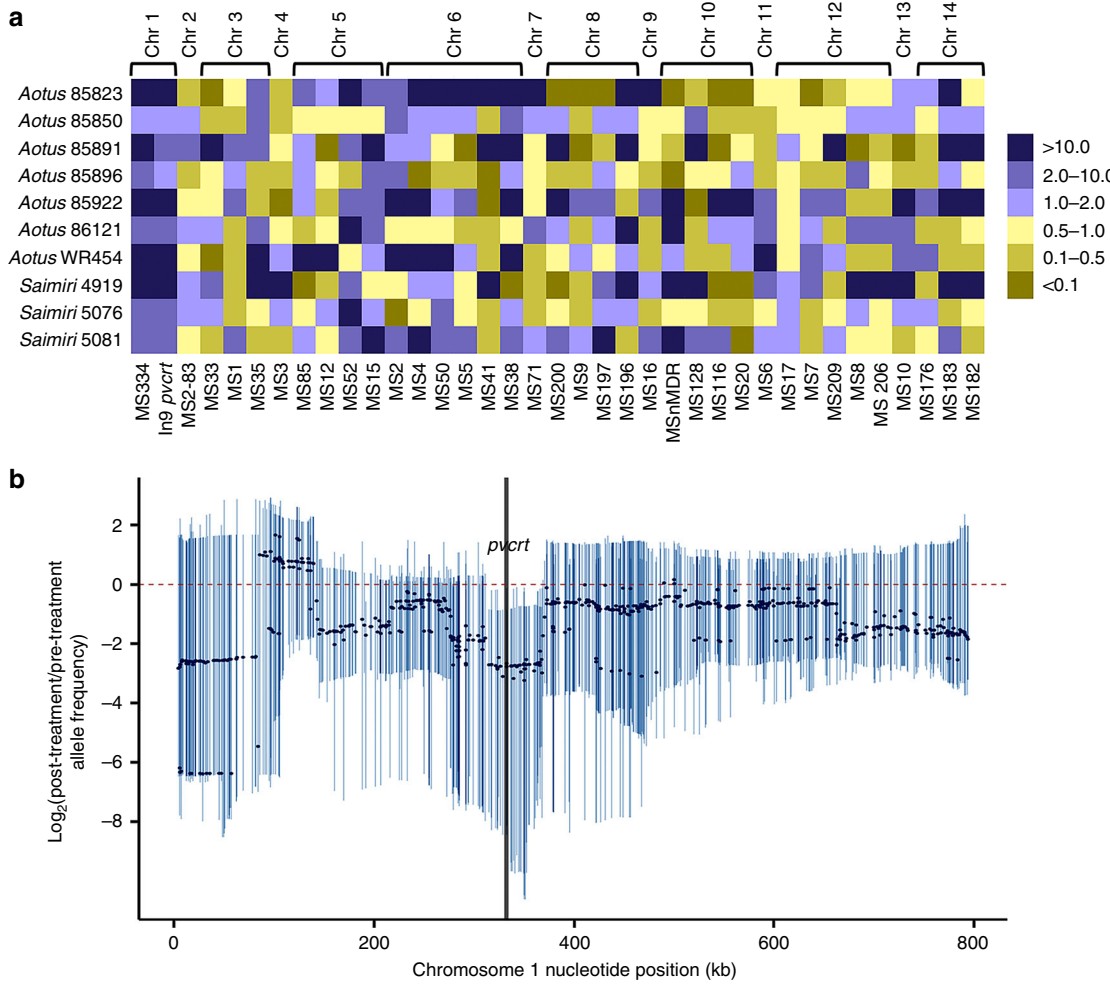

**Fig. 2** Effect of CQ selection on mixed progeny populations from the NIH-1993-R × NIH-1993-S cross. **a** Heat map showing relative prevalences of microsatellite polymorphisms among the mixed progeny populations before and after CQ selection in *Aotus* or *Saimiri* monkeys. Colored squares in the heat map represent $E$ values from microsatellite marker analysis of recrudescences in 10 monkeys (blue, $E > 1$; yellow, $E < 1$). Among the 14 *P. vivax* chromosomes, only a region of chromosome 1 shows positive selection in all samples after drug pressure (microsatellites MS334 and In9*pvcrt*). Columns with both blue and yellow squares indicate no consistent selection of any microsatellites in the other 13 chromosomes and are consistent with different combinations of individual progenies in each monkey. **b** Linkage group selection analysis of chromosome 1 by targeted genome sequencing. Aggregate smoothed log₂-fold changes of pre-CQ and post-CQ treatment allele frequencies are plotted as dots; 95% CIs for individual points based on bootstrapping with replacement, are indicated by the vertical lines. A bootstrap range that does not overlap with 0 suggests selection of the corresponding region with CQ exposure

showed increases ($E > 1$), as well as decreases ($E < 1$) among the LGS pairs for every marker except the two specific to chromosome 1: MS334 and a *pvcrt* intron 9 polymorphism (In9*pvcrt*), which showed positive selection in all LGS pairs in the post-CQ samples. MS334 is located ~1 kb upstream from the initiation codon of *pvcrt*. This finding of strong linkage to the NIH-1993-R phenotype is unlikely to be an artifact of redundant progeny pools, as each LGS recrudescence showed a distinct pattern of $E$ values consistent with a unique mixed population of recombinants (two-sided $p = 0.026$: under the null hypothesis that the markers are independently inherited between chromosomes and that their $E$-values are equally likely to be >1 or <1 without selection).

**A 76 kb region selected by CQ includes *pvcrt*.** To delimit the genomic region linked to CQR, we performed targeted genome sequencing[20,21] of chromosome 1 on DNA from six LGS pairs before and after CQ treatment (DNA was insufficient from four recrudescences detected only by PCR, despite use of whole genome amplification; Supplementary Fig. 4c, f, i, j). Using within-

sample allele frequencies smoothed by a 40-SNP running median sliding window in six monkey pairs (Supplementary Table 10), we identified a well-demarcated 76 kb selection valley syntenic with the region of *P. falciparum* chromosome 7 harboring the *pfcrt* gene[8] (Fig. 2b, Table 1). On chromosome 5, which contains a microsatellite marker (MS15) with $E$ values >1 in 8/10 recrudescent infections, targeted genome sequencing detected no region significantly selected by CQ (Supplementary Fig. 6; insufficient DNA was available to complete similar sequencing of the other chromosomes). We note that our search would not have been able to map the presence of mutation that was the same in both parents, nor does it dismiss the possibility that the 37 microsatellite markers could have missed the inheritance of separate, additional determinant required for resistance. Although the frequency of meiotic crossover events is not reported for *P. vivax*, information is available from two other *Plasmodium* species: *P. falciparum* and *Plasmodium chabaudi chabaudi*, a rodent malaria parasite that is evolutionarily closer to *P. vivax* than to *P. falciparum*[22]. Excluding the highly recombinogenic subtelomeric regions, genome genetic lengths of 1655

**Table 1 P. vivax genes in the 76 kb region of chromosome 1 selected by chloroquine in mixed populations of NIH-1993 S × R progeny**

| Gene ID | Location in chromosome 1 (P. vivax Salvador-I) | Product description |
| --- | --- | --- |
| PVX_087955 | 313,584–316,927 (−) | O1, putative |
| PVX_087960 | 317,375–319,103 (−) | Inner membrane complex protein 1d, putative |
| PVX_087970 | 324,282–326,945 (+) | Heat shock protein 110, putative |
| PVX_087980 | 330,260–334,540 (+) | Chloroquine resistance transporter, putative (pvcrt) |
| PVX_087985 | 337,343–340,573 (+) | Chloroquine resistance associated protein Cg1, putative |
| PVX_087990 | 341,247–342,248 (−) | Glutaredoxin domain containing protein |
| PVX_087995 | 342,633–349,964 (−) | Chloroquine resistance associated protein Cg2, putative |
| PVX_088000 | 350,982–355,492 (−) | Chloroquine resistance associated protein Cg7, putative |
| PVX_088005 | 356,035–356,651 (+) | Hypothetical protein, conserved |
| PVX_088007 | 357,334–361,250 (−) | rRNA/tRNA ribonuclease MRP/P subunit, putative |
| PVX_088010 | 362,843–365,107 (−) | PST-A protein |
| PVX_088015 | 366,877–368,031 (−) | Lysophospholipase, putative |

and 1570 centiMorgans (cM) have been reported for *P. falciparum*[23,24], and 1676 cM for *P. c. chabaudi*[25]. Another estimate of 2210 cM for *P. falciparum* did not distinguish the contributions of markers from subtelomeric regions including gene families involved in processes of antigenic variation[26]. For the purposes of our present discussion, use of core genetic lengths are appropriate for the 14 nuclear chromosomes, indicating 16.55, 15.70, and 16.76 crossovers per generation by definition of the centiMorgan. These yield averages of 1.19, 1.12, and 1.20 crossover per chromosome, but individual numbers vary with chromosome size[23–25]. We suggest that a particularly high level of recombination activity would have been necessary in the vicinity of a postulated second determinant for the markers to have missed it outside of the chromosome region carrying *pvcrt*.

**Differential *pvcrt* transcription associated with CQ response**. No non-synonymous changes of *pvcrt* were associated with CQR in the cross, consistent with previous findings[8]. To evaluate the possibility that changes in *pvcrt* expression confer CQR, we tested mRNA samples from five LGS pairs by quantitative reverse transcription-PCR (qRT-PCR). These experiments included eight samples of CQ-selected mixed progeny from the five LGS pairs. Results demonstrated greater *pvcrt* mRNA transcript levels in the CQ-selected mixed progeny (Fig. 3a, Supplementary Table 11; $2^{-\Delta\Delta Ct}$ ratio (CQ-selected/unselected) = 2.40 (95% CI: 1.53, 3.77; $p = 0.006$).

PvCRT protein quantitation can be difficult as it is a food vacuole membrane protein and material from blood samples or isolates under short-term ex vivo conditions is very limited. We therefore turned to targeted liquid chromatography multiple reaction monitoring with a heavy labeled (C13) isotopic peptide internal standard (LC–MRM–MS)[27] for its ability to quantify low abundance proteins from limited biological samples[28]. By this method, signals from three transitions were obtained from a peptide of PvCRT (Supplementary Table 12). Eighteen samples were obtained from 15 *P. vivax*-infected monkeys: four infected with mixed NIH-1993 S × R unselected progeny, four with CQ-selected mixed progeny, three with Chesson[29] CQ-S parasites, and two each with the CQ-R AMRU-I and Indonesia-XIX[30] lines. Parasite-infected red blood cells (pRBCs) were purified by a magnetic column and analyzed by LC–MRM–MS for total PvCRT content (Supplementary Tables 13 and 14). Calculations of cellular PvCRT content from these results yielded geometric means of 1.32 (95%CI, 0.45–3.91) and 2.62 (95%CI, 0.88–7.75) zeptomoles per pRBC in the unselected and selected NIH-1993 S × R progeny, respectively (Fig. 3b; two-sided Welch's *t*-test $p = 0.23$, geometric mean ratio 1.98, 95% CI: 0.58, 6.75). Wide confidence intervals of these estimates reflect the difficulties of

both low-level membrane protein measurements and recovery of similarly distributed *P. vivax* stages from ex vivo cultures. Comparable zeptomole amounts of PvCRT were measured in the pRBC of Chesson, AMRU-I, and Indonesia XIX lines (Supplementary Table 14).

**Polymorphisms in *pvcrt* non-coding regions**. Figure 3c shows the schematic of ~4.6 kb DNA sequence from NIH-1993 chromosome 1 carrying MS334 and the *pvcrt* gene. Sequence comparisons (Supplementary Fig. 7) identified multiple TGAAGH hexanucleotide units interspersed by 10 nucleotide (nt) blocks in two non-coding stretches of *pvcrt*: (1) Approximately 0.6 kb upstream from the start codon, where CQ-S progeny contain 10 TGAAGH motifs and CQ-R progeny contain 15 TGAAGH motifs in a region that differs by 82 nt; and (2) in intron 9, where CQ-S progeny contain 17 TGAAGH and CQ-R progeny contain 14 TGAAGH motifs in a region that differs by 60 nt. Interestingly, no analogous introns with TGAAGH motifs occur in the *crt* genes of other major *Plasmodia* clades, including those of *P. falciparum* and *P. knowlesi*; and *P. vivax* transcripts containing intron 9 sequences have been reported with a predicted early stop codon after the repeats[31]. Although there is currently no knowledge regarding their role in *Plasmodium*, TGAAGH motifs are involved in gene transcription regulation of *Arabidopsis*, including cis-acting regulatory elements or CAREs[32], intron-mediated enhancers[33], and splicing enhancer motifs[34] (Supplementary Table 15). Further evaluation of these and other possible regulatory elements of gene transcription activity in *P. vivax* will require experimental breakthroughs in culture and genetic manipulation.

**Discussion**

The *P. vivax* cross presented in this work links a CQR phenotype to a 76 kb region of chromosome 1 and greater expression of *pvcrt*, an ortholog of the *P. falciparum* CQR transporter gene. These results may help to explain evidence of selective pressure in the 5'UTR and intronic regions of *pvcrt* in previous studies[35–37], including occurrences in the top 1% of genes with intronic mutations, although others found no association between the *pvcrt* genomic region and *P. vivax* CQR[9,38]. Our evidence for upregulated *pvcrt* transcription is also consistent with some findings from patients with CQ-R vivax malaria[13,14]. However, such upregulation was not observed in ex vivo assays of patient samples[15]. To the extent that greater expression may reverse in the absence of drug pressure or not maintain well under laboratory conditions, *pvcrt* upregulation may be difficult to detect in ex vivo studies. We also note that determinants in addition to *pvcrt* may be involved in *P. vivax* CQR, or that strains

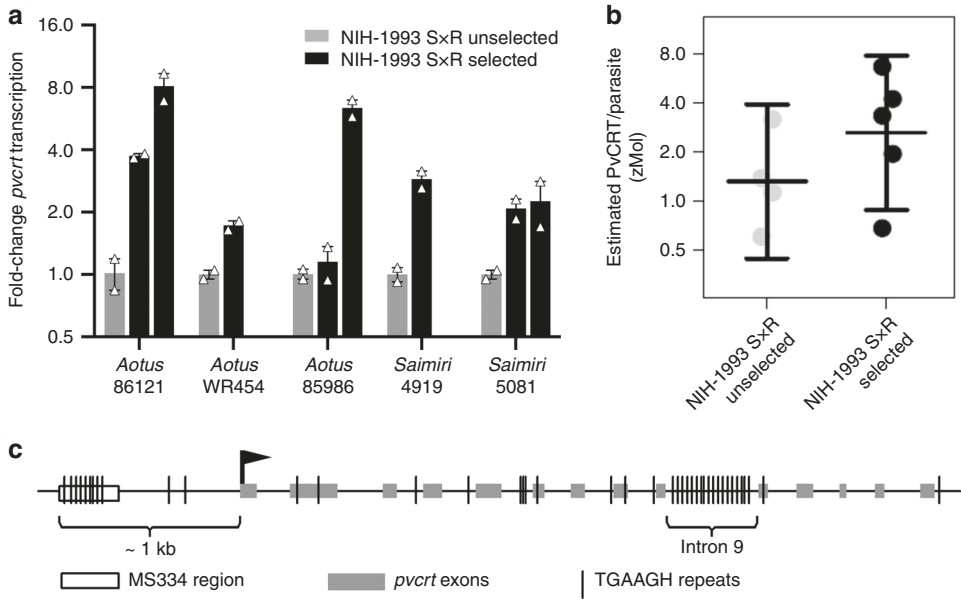

**Fig. 3** Evidence for increased *pvcrt* transcription in CQ-selected *P. vivax*. **a** Levels of *pvcrt* transcription in mixed progeny from the NIH-1993 S × R cross before and after CQ treatment in five different *Aotus* or *Saimiri* monkeys. Black bars indicate *pvcrt* transcription levels in the CQ-selected progeny, normalized to the transcription levels in unselected progeny before CQ treatment (gray bars). Transcription levels were calculated relative to those of the *pvseryl-tRNA synthetase* 'housekeeping' gene by the $2^{-\Delta\Delta cT}$ method[63]. PCR determinations for the calculations were performed in duplicate. Thawed samples of cryopreserved parasitized erythrocytes were used in all experiments, thereby enriching for ring-stage parasites and reducing potential variations from older parasite stages. A single post-CQ selection sample was collected from *Aotus* WR454 and from *Saimiri* 4919; post-CQ treatment samples were collected on two different days from each of *Aotus* 86121, *Aotus* 85986, and *Saimiri* 5081. Error bars indicate standard deviations from two technical replicates shown as white triangles in the graph. **b** Estimated PvCRT protein concentration per parasitized erythrocytes in monkey blood samples infected with from NIH-1993 SR unselected (gray) and selected (black) progeny. Points for each strain are the estimated levels, and lines represent geometric means and the associated 95% confidence intervals from 5 or 6 biologically independent samples. zMol zeptomole. **c** Schematic of the *P. vivax* chromosome 1 segment containing the exons, introns, and untranslated flanking regions of *pvcrt*. Multiple repeats of the TGAAGH motif occur in MS334 and intron 9. F9 indicates the region from exon 9 where the sequence of forward primer used to perform qRT-PCR is located and R11 the region where the sequence of the reverse primer is located (see the "Methods" section). The asterisk indicates where codons of the peptide used for MRM analysis are located (exon 13)

of *P. vivax* with higher levels of resistance may carry mutations not present in the NIH-1993-R × S parasites.

A fortuitous aspect of our study was the crossing of two distinct subpopulations from the NIH-1993 line, of which one was enriched by sub-therapeutic CQ treatment in the chimpanzee. *P. vivax* infections are commonly multiclonal[39,40], helping to explain why NIH-1993-S and NIH-1993-R are closely related to Sal-I in their genotypes, yet are distinct from each other and from Sal-I by markers including CSP types, *pvdhfr-ts* mutations, and microsatellites of *pvcrt*. Passages of Sal-I and NIH-1993 through different host systems, and pressures, such as drug exposure, would have caused the prevalence of various parasite subpopulations to change over time. Intriguingly, without CQ pressure, NIH-1993-R type parasites seem to have been less favored than NIH-1993-S in the monkey infections of our study. We speculate that the genetic changes *P. vivax* parasites need to survive CQ reduce their fitness relative to CQ-S parasites in the absence of drug pressure, a hypothesis that might help to explain the slow rise and spread of CQ-R *P. vivax* relative to *P. falciparum* in malaria-endemic regions.

The evidence for *pvcrt* upregulation in CQ-R *P. vivax*, as well as previously demonstrated effects of PvCRT in heterologous systems[11,12] raise intriguing questions about the native function of this transporter and its role in drug resistance. Why is PvCRT able to transport CQ, whereas wild-type PfCRT in *P. falciparum* can do so only when it harbors specific amino acid mutations[12]? How might these various observations relate to findings of different stage-specific drug responses of *P. vivax* and *P.*

*falciparum*[31,41,42] or to possible effects on resistance from post transcriptional modifications of the protein[43]? How do PvCRT and PfCRT differ in their molecular functions? In view of these questions, it is interesting to consider a proposed model in which the mutations of *P. falciparum* CQR might convert the carrier function of PfCRT from an exchange-only activity to a net transport function for the drug[44]. In the case of PvCRT, net transport of CQ may already be an inherent directional property of the transporter. Structural explanations for the different properties of PfCRT and PvCRT will have broad implications for our understanding of the CRT family of carriers, the natural functions of which remain unknown.

## Methods

**Plasmodium lines**. *P. vivax* NIH-1993 was obtained from NIH archive cryopreserved samples from *Aotus* T308, May 1993. The following strains were obtained from the Biodefense and Emerging Infections Research Resources Repository (BEI resources, NIAID, NIH, Manassas, VA, USA): *P. vivax* AMRU-I (catalog number MRA-372), Indonesia-XIX (catalog number MRA-378), and Chesson (catalog number MRA-383), contributed by William E. Collins. Parasite line NIH-1993 line was initially identified in NIH stocks as the CQ-S Salvador-I strain. However, comparisons to the nuclear chromosome sequences and the mitochondrion genome of Salvador-I [https://www.ncbi.nlm.nih.gov/nuccore/AY598140.1] showed polymorphisms that distinguish NIH-1993 and Salvador-I, including three dihydrofolate reductase mutations at codon positions 57, 99, and 173 (Supplementary Table 1).

***P. vivax* nonhuman primate infections**. All animal procedures were performed in compliance with the National Institutes of Health (NIH) Guidelines under protocols approved by the National Institute of Allergy and Infectious Diseases Animal

Care and Use Committees (NIAID ACUC) and from Bioqual Inc. (Rockville, MD, USA). *Aotus nancymaae*, *Saimiri boliviensis*, and *Pan troglodytes* (hereafter referred to as *Aotus*, *Saimiri*, and chimpanzee) were obtained from NIH approved sources and housed in compliance with the Animal Welfare Act and the Guide for the Care and Use of Laboratory Animals[45]. Cryopreserved or fresh NHP-parasitized blood samples (between 0.5 and 1.5 mL) were thawed and washed in sterile RPMI 1640 (KD Medical, Columbia, MD, USA) and inoculated intravenously (IV) into animals anesthetized with 10 mg/kg ketamine. Thin and thick blood smears from animals inoculated with *P. vivax* were checked weekly until parasites were detected microscopically (in counts of ~10,000 erythrocytes from a thin blood smear, fixed with methanol and stained with 20% Giemsa), then checked daily. For blood smears and hematocrit measures, between 0.01 and 0.05 mL blood were obtained from the saphenous vein or by ear prick; larger blood draws for parasite cryo-preservation, DNA/RNA extraction, and drug plasma level measurements were obtained from the femoral vein of anesthetized animals.

Treatment regimens of 3 days of 2.5, 5.0, or 10.0 mg/kg/day CQ (delivered via gavage) were administered to monkeys that achieved a minimum parasitemia of 0.2%. A blood sample of ~1.5 mL for parasite cryopreservation, DNA, and plasma analysis was collected prior to the first dose of treatment. Another 0.3–0.5 mL blood sample was collected for CQ plasma measurement on the 4th day after completion of treatment. Parasitemias were monitored daily for 7 days from the beginning of CQ treatment or until parasitemia cleared and then were checked for parasite recrudescence at least weekly until 62 days after treatment. Blood samples from the recrudescences were collected for cryopreservation and other analyses indicated in the text.

Definitive cure of New World monkeys infected by blood stage parasite inoculation was achieved by mefloquine HCl (single 25 mg dose) or by three daily administrations of 25 mg atovaquone/10 mg proguanil HCl (Malarone, pediatric weight-adjusted dose). Cure of the chimpanzee infection from the blood-stage inoculation was by three daily administrations of 187.5 mg atovaquone/75 mg proguanil HCl (Malarone, pediatric weight-adjusted dose); cure of the infection from the sporozoite inoculation in the more grown chimpanzee was by three daily administrations of 250 mg atovaquone/100 mg proguanil HCl (Malarone, adult strength), plus primaquine base to eliminate liver stage hypnozoites (3 days of 34 mg/day)[46]. The antimalarials were delivered orally from preparations of crushed tablets mixed in Ensure liquid. Clearance of any residual infection was confirmed by a high sensitivity PCR test[47].

**P. vivax mosquito infections and sporozoite inoculations**. For production of *P. vivax* sporozoites, 3–5-day-old female *A. stephensi*, *A. freeborni*, *A. dirus*, and *A. gambiae* mosquitoes were raised in the LMVR/NIAID and at Sanaria Inc. Individual secure pint or gallon containers were populated with up to 100 and 500 female mosquitoes, respectively. After starvation of 24 h, the mosquitoes were fed for 10–20 min via artificial membrane feeding or directly onto the shaved abdomen of the anesthetized chimpanzee. The fed mosquitoes were maintained with 5% corn syrup and water at 26 °C for up to 18 days in a secure insectary. Randomly selected mosquitoes were dissected 6–9 days post-feeding and examined for oocysts in the midgut; additional mosquitoes were randomly selected and dissected after 12–14 days to check for sporozoites in the salivary glands. To assess oocyst presence and numbers, midguts from 10 to 15 dissected mosquitoes/container were stained with 0.5% mercurochrome for 10 min then observed with a microscope using a ×20 or ×40 objective lens. Stained oocysts were isolated from the midgut using #5 tweezers (Sigma, St. Louis, MO, USA) and a 23-gauge needle attached to a syringe containing phosphate-buffered saline (PBS; 10 mM Na$_2$HPO$_4$, 2 mM KH$_2$PO$_4$, 137 mM NaCl, 2.7 mM KCl, pH 7.4). Individual pieces of mosquito midguts containing single oocysts were transferred to 50 µL of a homogenization buffer containing 80 mM EDTA, 100 mM Tris pH 8.0, and 160 mM sucrose[48], and frozen in 1.5 mL tubes at −20 °C until DNA extraction. Sporozoites were isolated, purified, cryopreserved, and confirmed to be infective in hepatocyte invasion assays[18]. For inoculation of the chimpanzee with progeny of the cross, cryopreserved sporozoites were rapidly thawed in a 0.6% albumin saline solution suspension for 30 s in a circulating water bath at 37 °C and inoculated intravenously within 15–30 min of thawing.

**Immunofluorescence antibody assay (IFA) of sporozoites**. Purified sporozoites were suspended in PBS containing 2% bovine serum albumin (BSA, Life Technologies, Carlsbad, CA, USA). Twenty microliters of this suspension containing ~2000 sporozoites were spotted into individual wells of an immunofluorescence slide (Erie Scientific, Ramsey, MN, USA). The slide was air-dried at room temperature for 24 h before IFA of the *P. vivax* circumsporozoite protein (PvCSP) with monoclonal antibody (MAb) VK210, which recognizes the PvCSP common repeat GDRADGQPA (Mab NVS3 Cat#WRAIR, gift from CDC Dr. Robert Wirtz) and MAb VK247, which recognizes the PvCSP variant repeat ANGAGNQPG (Cat#MRA-1028K, BEI Resources, NIAID, NIH, Manassas, VA). Twenty microliters of MAb VK210 (used at 0.1 µg/mL, which is a 1:10,000 dilution of the stock of 1 mg/mL) was added to each slide well containing air-dried sporozoites and incubated at 37 °C in a humid chamber for 30 min. After three washes of the slide with PBS (5 min each), 20 µL of Alexa Fluor 488 Goat anti-mouse IgG (Product # A28175, Thermo Fisher, used at 1 µg/mL in phosphate-buffered saline containing 0.2% BSA) were added to each well and incubated for 30 min at 37 °C in a humid

chamber. After three washes again with PBS, 20 µL of PvCSP VK247 MAb at 0.1 µg/mL were added to each well. Slides were then incubated at 37 °C in a humid chamber for 30 min, washed three times with PBS, and processed with 20 µL of Alexa Fluor 568 Goat anti-mouse IgG (Product # A-11004, Thermo Fisher, used at 1 µg/mL) as described above for Alexa Fluor 488. Cover slips were applied to the slides using Vectashield mounting medium (Vector Laboratories, Inc., CA, USA). Examination and imaging was performed in an Olympus BX-40 fluorescent microscope equipped with FITC and TxRed filters.

**Mature blood-stage parasite isolation and DNA extraction**. Magnetic LS separation columns (MACS®, Miltenyi Biotec, Sunnyvale, CA, USA) were used to isolate mature blood-stage parasites from uninfected erythrocytes, host white blood cells, and platelets[49]. NHP blood samples were centrifuged at 2500 rpm for 5 min to separate and remove serum. Blood cells were used to make a 0.5–1.5% hematocrit solution in RPMI 1640 (KD Medical). The magnetically purified parasites were eluted with 3 mL RPMI 1640, split into two 1.5 mL tubes. After centrifugation at 2500 rpm for 5 min, ~90% of the volume of each tube was discarded and the pellet was frozen at −20 °C. DNA from these cells was obtained by standard phenol–chloroform methods[50], suspended in 50 µL of 10 mM Tris–0.1 mM EDTA pH 8.0, and frozen. These same methods were used to extract DNA from oocysts and sporozoites. To increase DNA template for microsatellite analysis or genome sequencing, whole genome amplification (WGA) was performed with 2–5 µL of samples of extracted DNA using the Repli-G midi kit (Qiagen, Germantown, MD, USA)[51].

**RNA isolation and cDNA synthesis**. Cryopreserved samples of parasitized blood were thawed by standard methods, which favor the recovery of ring-stage infected erythrocytes[52]. RNA was extracted by phenol–chloroform methods[50] after solubilization of the thawed cells in Trizol (Sigma). Samples of 1–3 µg total RNA were treated with Amplification Grade Deoxyribonuclease I as recommended by the manufacturer (Life Technologies); cDNA was synthesized with the SuperScript III First-Strand Synthesis Super Mix kit using random primers (Life Technologies).

**P. vivax microsatellite analysis**. All genomic DNA samples were genotyped and analyzed blindly. A list containing the sequence of all oligonucleotide primers used in this study is presented at the end of the Methods section. Thirty-five polymorphic microsatellite markers that distinguish the NIH-1993-S, NIH-1993-R, and AMRU-I parasites were identified from a screen of 232 previously reported microsatellites sequences distributed over the 14 *P. vivax* chromosomes[53–56] (Supplementary Table 9). Because only 35 markers of the 232 tested differed between NIH-1993 S and R, two additional microsatellite markers were identified to carry polymorphic regions in *pvcrt* intron 9 (marker In9*pvcrt*) on chromosome 1, and in a sequence nearby *pvmdr-1* (MSnMDR) on chromosome 10. Between 20 and 100 ng of WGA DNA were amplified using Platinum PCR SuperMix High Fidelity polymerase (Life Technologies) and a 6-fluorescein amidite-modified fluorescently labeled oligonucleotide in the reverse primer of each pair (0.5 µM). Amplified PCR products were obtained after 42 cycles of 20 s at 94 °C, 10 s at 52 °C, 10 s at 47 °C, and 30 s at 60 °C. The PCR products were diluted 100–1000-fold and evaluated for size differences and relative intensities using an ABI 3730xl DNA Analyzer (SeqGen, Torrance, CA, USA). The CQ selection effect *E*-value of each marker was calculated as the 'ratio of ratios' of signals from the two inherited parental polymorphisms in LGS progeny pairs after and before CQ treatment:

$$E = \left(\mathrm{marker}_{\mathrm{NIH-1993-R-after}}/\mathrm{marker}_{\mathrm{NIH-1993-S-after}}\right)/\left(\mathrm{marker}_{\mathrm{NIH-1993-R-before}}/\mathrm{marker}_{\mathrm{NIH-1993-S-before}}\right)$$

**Targeted Illumina sequencing**. DNA of MACS-purified parasites from the 10 LGS pairs pre-CQ and post-CQ treatment was whole genome-amplified as described above. Sample-indexed shotgun Illumina TruSeq libraries were generated from WGA DNA by recommended kit conditions (Illumina, San Diego, CA, USA). Of the 10 LGS pairs, six (from *Aotus* 85922, 85986, 85850, WR454, 85823, and *Saimiri* 4919) were fully analyzed. Results from four LGS sample pairs could not be included because recovered DNA was not sufficient from the PCR-positive-only recrudescences in the monkeys (*Aotus* 86121, 85891, *Saimiri* 5081, and 5076; Supplementary Fig. 4). Chromosomes 1 and 5 sheared fragments were captured using custom SureSelect RNA baits with 3 × tiling (Agilent Technologies, Santa Clara, CA, USA) and were sequenced on an Illumina HiSeq2000.

Reads were aligned to the *P. vivax* Salvador I reference assembly genome (version 3) using bwa-mem with default settings. Aligned reads were deduplicated with Picard Tools MarkDuplicates and then had mate information fixed with Picard Tools FixMateInformation. Finally, the aligned reads underwent local realignment of entropic regions using Genome Analysis Toolkit (GATK) IndelRealigner[57–60]. Read coverage and statistics were calculated with SAMtools flagstat[61] and custom R scripts.

Variant discovery was performed using samtools mpileups with flags for max-depth of 10,000 bases, minimum base quality of 20, and a minimum mapping quality of 10. The output from mpileups was then passed to bcftools multiallelic-caller. Loci were filtered using bcftools filter for an INFO field mapping quality of <55 and if a SNP occurred within five-base pairs of an INDEL. Loci were also excluded if they were not within the core region defined by Pearson, Amato[9] or if

they were annotated in the *P. vivax* Salvador I GFF (version 3) as members of the *msp*, *Pvfam-c*, *VIR*, *SERA*, or *STP* family[21]. In addition, variants were subsetted to biallelic SNPs. These were filtered further and excluded if the within-sample allele depth was <10 or if the phred-scaled strand bias *p*-value was >10. After these filters were applied, variants were excluded across all samples if more than 2/12 samples had missing information or if the range of within-sample allele frequencies across all samples was <5% suggestive of genotyping error.

**Quantitative real time PCR (qRT-PCR) analysis of *pvcrt*.** All qRT-PCR analyses were performed using the Rotor-Gene SYBR Green PCR Kit according to manufacturer's instructions (Qiagen). For each reaction a total of 30 ng cDNA and 1 μM of each primer were mixed with the Rotor-Gene PCR SYBR Green mix and analyzed in the Rotor-Gene Q over 35 cycles, each consisting of two steps: 10 s at 96 °C and 30 s at 60 °C. For quantitative transcription analysis, the threshold cycle value (Ct) was used to estimate the ΔCt of each sample relative to that of the house keeping gene seryl tRNA synthetase (PVX_123480, an ortholog of *P. falciparum* PF3D7_1216000);[62] calculated 2$^{-\Delta\Delta Ct}$ values[63] provided comparative measurements of *pvcrt* transcript levels in samples from each LGS pair before and after CQ selection. Expected PCR products from cDNA were 151 bp for *pvcrt* using oligonucleotide primers pvcrt-RT-Fwd and pvcrt-RT-Rev from exon 9 to 11, and 116 bp for *pvseryl-tRNA-synthetase*, using primers pvSerylT-RNA-Fwd and pvSerylT-RNA-Rev. A list containing the sequences of oligonucleotide primers used in this study is presented in Supplementary Table 16.

**Quantification of PvCRT protein levels.** We performed liquid chromatography multiple reaction monitoring (LC–MRM) using a triple quadrupole (QQQ) mass spectrometer to measure the abundance of PvCRT in infected NHP erythrocytes infected with various strains of *P. vivax*. For this purpose, we identified candidate tryptic peptides likely to be of high quality, and which confirmed with the minimum general rules for the peptide's use in LC–MRM: no fewer than eight amino acids; no known post-translational modifications; no missed cleavages; and no predilection to artifactual modifications (empirically determined by MS/MS analyses). Before use of the limited *P. vivax* samples, we performed experiments with the orthologous PfCRT peptides from *P. falciparum* cultures, which provided larger amounts of infected red cells and samples of greater scale. An orthologous peptide of PvCRT IGNIILEK (the PfCRT peptide VGNIILER; 85% sequence identity) met the quality thresholds for confident identifications and was readily identified in replicate LC–MS/MS runs. Another PfCRT peptide identified from our in silico predictions could also be observed, but it was less abundant, supporting VGNIILER as the preferred choice for PfCRT quantification in samples with low parasite densities. By procedures described below, testing of samples from *P. vivax*-infected *Saimiri* confirmed that the orthologous PvCRT peptide, IGNIILEK, was readily detected at low parasitemias and provided the requisite high-quality spectrum for PvCRT quantification.

To reduce the number of uninfected cells, which contribute to hemoglobin contamination, and enrich parasite blood stages with a developed food vacuole, where PvCRT is expected to be expressed, each blood sample was passed through a MACS column. The number of parasites to be purified was estimated based on the percentage parasitemia of hemozoin-containing parasites (trophozoites, schizonts, and gametocytes), and number of cells that passed through the column was counted using a cell counter (Nexcelom Biosciences, MD, USA). The eluate was treated with saponin washed with SigmaFast$^{TM}$ protease inhibitor (Sigma) and snap frozen. After thawing, samples were suspended and boiled at 95 °C in variable volumes of SDS and dithiothreitol buffer (4% (w/v) SDS, 0.1 M Tris/HCl pH 7.6, 0.1 M DTT) as listed in Supplementary Table 14. Part of the extracted protein supernatant was digested using the Filter Aided Sample Prep (FASP) procedure[64,65] (volumes listed in Supplementary Table 14) and HPLC was performed to desalt the FASP-digests. Prior to LC–MRM analysis, the sample load was estimated and normalized by comparison with soluble samples whose concentrations, as determined by bicinchoninic acid (BCA) assay, were known. This approach allowed us to estimate the concentration of the FASP-digestion peptide products at the HPLC desalting step by using the integrated area at 214 nm and standardizing to the known concentration of total digested peptides.

We obtained MRM determinations that included three transitions for the selected PvCRT IGNIILEK peptide (Supplementary Table 12); the choice of three pair-transitions detected a 6 Da difference between an internal $^{13}$C heavy-labeled lysine standard peptide and the digested PvCRT peptide product ions. Desalted peptide digests (~30 μg) were spiked with 100 fmol heavy labeled lysine (K) internal standard peptide IGNILEK($^{13}C_6$). An Agilent 1290 Infinity UHPLC system was used to inject 17 μL of the desalted peptide digests/internal standard mix onto a reversed phase analytical column (150 mm × 2.1 mm i.d., Agilent Poroshell 120 EC-C18, 2.7 μm particle size) that was maintained at a temperature of 40 °C. Elution of peptides from the analytical column was performed using a gradient starting at 97% A (A: 99.9% water, 0.1% formic acid (FA)) at 0.5 mL/min. The mobile phase was 3–10% B (B: 90% acetonitrile (ACN), 9.9% water, 0.1% FA) for 3 min, 10–45% B for 9 min, 45–99% for 1 min, and maintained at 99% B for 3 min, followed by re-equilibration of the column with 3% B for 4 min. An Agilent 6490 QQQ LC/MS with iFunnel technology, controlled by Agilent's MassHunter Workstation software (version B.06.00), was used for all LC–MRM sample analyses. All acquisition methods used the following parameters: 3000 V capillary

voltage and a 1500 V nozzle voltage, a sheath gas flow of 11 L/min (nitrogen) at a temperature of 250 °C, a drying gas flow of 14 L/min at a temperature of 200 °C, nebulizer gas flow at 20 psi, and Q1 and Q3 set to unit resolution (0.7 FWHM). MRM acquisition methods were composed of three ion pairs during the acquisition of high-signal producing interference-free transitions and LC method development. A default 380 V fragmenting voltage and 5 V cell accelerator potential were used for all MRM ion pairs, and the dynamic MRM option was used for all data acquisition with a target cycle time of 500 ms. The collision energy for both the heavy labeled and light peptides was set at 20.0 V.

**Statistical analysis.** *Analysis of Fig. 2a*: In Fig. 2a, if there was no change in the prevalence of the NIH-1993-R and NIH-1993-S between the before and after CQ treatment, then we would expect that the *E* value for each monkey and marker would be equally likely to be >1 or <1. We test this null hypothesis using a Monte Carlo version of a within-cluster resampling test[66]. To motivate this, first note that there are 37 polymorphic microsatellite markers measured on 14 chromosomes. Because *E* values are likely to be correlated within chromosomes, we cannot assume the 37 markers are independent, instead we assume independence between markers on different chromosomes. Under this assumption, we can randomly select one marker from each chromosome and test for an effect on those 14 markers. The *p*-value is the simulated probability of observing equal or more extreme data (for Fig. 2a, this is the probability that 1 or more out of the 14 chromosomes will have all 10 *E* values either > 1 or < 1). The resulting simulated two-sided *p*-value (using 1e + 5 replications) is $p_1^{(\text{chrom})} = 0.02624$. This *p*-value applies to any random selection of 14 markers (one marker from each chromosome) for the data in Fig. 2a, since the only markers with all 10 *E* values the same are both on chromosome 1. Because the *p*-value could apply to any random selection, the *p*-value can be interpreted as a within-cluster resampling test of the entire figure[66].

*Analysis of Fig. 2b and Supplementary Fig. 6*: To detect selection valleys, we calculated within-sample allele frequencies (WSAF) from the filtered SNP-sites to produce a pre-CQ and post-CQ treatment WSAF for each monkey. Resulting WSAF were then smoothed with respect to chromosome and position by using a 40-SNP running median sliding window with the "median" endrule and the Turlach algorithm from the R-statistical software `stats` package [https://www.R-project.org]. As a result of this smoothing algorithm, samples with missing WSAF were dropped from the sliding window, the first 20 SNPs on each chromosome were burned-in, and the last sliding-window on each chromosomes was a subset of the 40-SNPs, as chromosomes are uneven. Correlations due to linkage disequilibrium were not taken into account, as we were searching for signs of linkage-disequilibrium with the linkage-group selection method[19].

In order to aggregate the smoothed-WSAF, we performed 10,000 bootstrap iterations (selecting six monkeys with paired pre-CQ and post-CQ treatment smoothed-WSAF with replacement) and calculated the median of the smoothed-WSAF at each of the loci for each iteration with respect to treatment status. From these bootstrapped results, we then performed a log-base-2 fold transformation between the pre-CQ and post-CQ treatment smoothed-WSAF. As a result of this transformation, decreases in smoothed-WSAF from pre-CQ to post-CQ treatment will be expressed as a negative value, while the null value of zero will indicate no change in in smoothed-WSAF with respect to treatment status. To account for allele frequencies of 0% we added a 1e−3 tolerance to all log-base-2 calculations. This tolerance factor is the lower bound of the confidence intervals. The median and the 95% confidence interval from the bootstrapped results are presented in Fig. 2b and Supplementary Fig. 6. Given that there were so few samples, the 95% confidence intervals are rough approximations. Further, there was no adjustment for multiplicity. Nevertheless, the intervals may be treated as descriptive statistics, so intervals that do not cross the null value of 0 to highlight loci where the smoothed-WSAF decreased between pre-CQ and post-CQ treatments. These regions likely indicate linkage group selection valleys.

*Analysis of Fig. 3a*: The panel shows data from five monkeys: two measurements (from unselected and selected progeny) are provided from each of two monkeys, and three measurements (one from unselected and two from selected progeny) from each of three monkeys. If each monkey had been measured only once after CQ treatment, then we could take the log of the fold change for all five monkeys and perform a *t*-test. Taking the mean of the log(fold-change) and exponentiating gives the geometric mean of the fold-change, and exponentiating the confidence limits gives a confidence interval on the fold-change. Since we repeated after-treatment measurements on three monkeys, we use within-cluster resampling to perform the analysis[66]. Similar to the *t*-test method described above, we work on the log-transformed responses and back-transform the results. Instead of using a normal distribution for inferences, we use the t distribution with four degrees of freedom, so that if there had been no repeated measurements, the results would be equivalent to the previously mentioned *t*-test methods.

*Analysis of Fig. 3b*: Confidence intervals used log-transformed values and either a one-sample *t*-test (for within group confidence intervals) or Welch's two-sample *t*-test (for comparisons between groups). Resulting CIs are exponentiated to return to the original scale (one-sample CIs) or to give CIs in terms of ratios of geometric means (two-sample CIs).

**Reporting summary.** Further information on research design is available in the Nature Research Reporting Summary linked to this article.

## Data availability

Genome sequencing data from the NIH1993-S parent are deposited in the NCBI Short Read Archive under accession number SRR2148599. Targeted Illumina sequencing data of NIH-1993 S × R progeny LGS sample pairs are deposited under accession numbers SRR2148595, SRR2148596, SRR2148597, SRR2148598, SRR2148600, SRR2148601, SRR2148602, SRR2148603, SRR2148604, SRR2148605, SRR2148606, and SRR2148607. The data underlying Fig. 1a, Supplementary Fig. 2a, and Supplementary Table 5 are provided as a Mendeley dataset [https://data.mendeley.com/], Sa, Juliana (2019), "Pv-X raw data", https://doi.org/10.17632/4455hbjb9g.1.

## Code availability

The analyses for Figs. 2a, 3a, b were performed using the R computer language and the Sweave system of Rstudio [http://www.rstudio.com]. The files are provided as Supplementary Software: Analyses_for_Figures_2a_3a_3b.Rnw (Sweave file) and the associated PDF file. R code employed for the analyses is embedded in the.Rnw file to create the.pdf file.

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

## Acknowledgements
We thank Ahlin Bruce, Billeta Lewis, Faith Sentz, Sekou Savane, Andy Limeric, Kevin Ham, John Bacher, Randy Elkins, Anthony Cook, Sali Muhammad, Zac Pippin, Bharathi Avula, Andre Laughinghouse, and Kevin Lee for their efforts in support of this project. This study was supported by the Intramural Research Program of the National Institute of Allergy and Infectious Diseases, National Institutes of Health; also funded in part by an 1R21AI111108 award to J.J.J., a seed grant from the Bloomberg Family Foundation to R.R.D., and T32 grants GM008719 and GM007092 to C.M.P.

## Author contributions
J.M.S., S.R.K., R.R.M.B., R.E.S., V.M.-M., W.A.K., P.K.H., E.G.K., S.O.-G., L.E.L., T.E., and M.L.T. performed nonhuman primate experiments; J.M.S., S.R.K., E.G.K., Y.A., E.R. J., S.V., and S.C. performed *Anopheles* mosquito experiments; E.R.J., cryopreserved sporozoites; S.V. performed hepatocyte invasion assays and sporozoite immuno-fluorescent images; J.M.S. performed microsatellite linkage analysis; N.F.B., C.M.P., D.K. D., J.A.B., and J.J.J. performed genome sequence and linkage analysis; P.S.F. and L.W. performed chloroquine measurements in nonhuman primate plasma; D.T. and R.R.D. performed proteomic experiments and analysis; J.M.S., S.R.K., R.R.M.B., R.E.S., T.J.G., V. M.-M., W.A.K., R.T.E., M.A.K., A.K., D.S., and J.M. performed molecular analysis of *pvcrt*; M.P.F. performed statistical analysis; B.K.L., S.L.H., and R.W.G. directed *Anopheles* mosquito production; J.M.S. and T.E.W. designed experiments, analyzed data, and wrote the manuscript; all authors reviewed and approved the manuscript. S.R.K., R.R.M.B., R.E. S., N.F.B., J.J.J., D.T., R.R.D., M.A.K., A.K., and D.S. contributed to the manuscript preparation.

## Additional information

**Competing interests:** The authors declare no competing interests.

