## [Peer Review File · Nature Communications]

Reviewers' Comments:

Reviewer #1:

Remarks to the Author:

This manuscript by Sa et al. documents a quite remarkable investigation to identify a marker of chloroquine (CQ) resistance in the human malaria parasite *Plasmodium vivax*. This has been a long sought after goal and here the authors tackle this by implementing a genetic cross between *P. vivax* parasites in a splenectomized chimpanzee. Because *P. vivax* blood stage parasites cannot be cultured in vitro, the authors use CQ pressure and linkage group selection to scan the genome in the search for a region that is positively selected in pools of CQ-selected progeny (passaged in *Aotus* or *Saimiri* monkeys). Using a panel of 37 microsatellite markers, they identify enrichment of a chromosome 1 region containing the *P. vivax crt* (*pv crt*) ortholog of *pf crt*, the CQ resistance determinant in *P. falciparum*. RT-PCR and protein expression data support increased expression of *pv crt* in CQ-selected progeny. Potential DNA changes in non-coding regions that might be causal for altered levels of gene expression are identified. One complicating and significant twist in this study came with the finding that the original CQ-resistant AMRU strain did not propagate in the chimpanzee, and that the two parents were separate isolates existing in a mixed isolate NIH-1993, of which one was enriched by sub-therapeutic CQ treatment.

The work behind this study is colossal and the authors deserve considerable credit for tackling this very complex project. I believe that this article meets the level of interest and scientific originality and robustness that merits publication in *Nature Communications*. There are however several qualifications that I believe must be addressed in a revised manuscript.

One major comment is that the authors employ only 37 microsatellite markers across the *P. vivax* genome, comprised of 14 chromosomes. There is therefore a reasonable chance that this low marker density would not have detected other chromosomal regions that would have shown positive selection in the CQ-treated progeny pools. This caveat should be clearly stated in the discussion and the number of markers mentioned in the abstract.

Another major comment concerns their estimation of PvCRT expression levels based on one peptide (IGNILEK). Were no other peptides from this protein identified or where those data too variable? This needs to be clarified. I would have thought that predicting differences in protein expression levels based on a single peptide is prone to inaccuracy.

Also, given that the authors could enrich for the CQ-resistant subpopulation with CQ treatment, did they obtain a pure resistant population? Was it possible to generate whole-genome sequence data from that parent? If so, those data should be deposited. Is the genome sequence data from the sensitive parent too heterogeneous (because of the resistant subpopulation) to report?

Minor comments:

Line 140. The authors should clarify in the text that the MS334 and *pv crt* intron 9 markers show positive selection in all LGA pairs in the post-CQ samples.

Line 192. Should be "breakthroughs in culture"

Line 229 is awkwardly worded: "parental lines blood-stage".

Figure 2b needs a scale on the X axis.

Figure 3 focuses on TGAAGH repeats, as does the text. Evidence that this might be significant comes from studies in other systems. Is there any evidence that this element might be important in any *Plasmodium* species? If not then that should be stated. It is clear that such an experiment cannot be performed in *P. vivax* at present.

Figure 3a also needs numbers of independent and technical repeats listed.

Line 490. I think this should read 10 chromosomes, not 14.

Figure S2: I would indicate that the Malarone treatment was over multiple days.

Reviewer #2:

Remarks to the Author:

The manuscript by Sa and colleagues attempts to address an important knowledge gap concerning *Plasmodium vivax* malaria – specifically, determining the molecular determinant(s) of chloroquine resistance (CQR) in this species. An undoubtedly huge amount of work has gone into the study and the authors have made a concerted effort to pull together experts from various disciplines to aid in different aspects of the experimental, laboratory and data analysis components of the study. There are unfortunately still some limitations that somewhat constrain the confidence of the results. Further discussion/acknowledgment of these limitations (described further below) is needed to give readers more clarity in interpreting the data.

1. The most substantial limitation is the initial inability to generate progeny between the intended parental lines, AMRU1(CQR) and NIH-1193 (CQS). As a result, the project appears to be constrained from the start to investigating the progeny of a more accidental cross using the CQS NIH-1993 strain (NIH-1993-S) and a derived strain, which the authors claim is CQR (NIH-1993-R). Unless I'm mistaken, it appears that the initial basis for defining NIH-1993-R as CQR (and for going ahead with the cross) appears to be that this strain was not detected until after the host was subject to a sub-therapeutic dose of CQ. However, it should be acknowledged that this is only an assumption. The possibility remains that the NIH-1993-R strain is CQS and simply took longer to amplify owing to a much lower initial parasite density and/or other growth-related phenotypic variation relative to NIH-1993-S. Indeed, the CQS sensitive NIH-1993-S strain appears to have been able to persist through the sub-therapeutic CQ dose fairly well. It would be helpful for the authors to discuss the aforementioned limitations further to give the reader more clarity on the potential issues.

2. Another limitation of the study is the limited (and somewhat biased) representation of the genome by the microsatellite marker set used in the linkage group selection analysis. While some chromosomes have several markers, others (chrs 2,4,7,9 and 11) are only represented by a single marker – this would surely constrain the ability to detect signals on those chromosomes. How were the 37 microsatellites selected? The authors might also consider adding a figure showing the genomic locations of the markers.

3. Following on from comment 2, it is a shame that the authors restricted the genomic analysis to chromosomes 1 and 5. Is this because there wasn't sufficient DNA even though they used WGA? For clarity for the readers, the authors might consider openly noting this is a limitation requiring further exploration in future studies.

4. Why were different hosts (some Aotus and some Saimiri) used in the drug response analysis?

5. Why did the total dose of CQ used in the drug response analysis vary (from 15-55 mg/kg total)?

6. I appreciate the authors' note on line 371 that cryopreserved samples were thawed by methods that should in theory favour ring stages, however, was this confirmed with microscopy data? If so, was the data on the percentage of rings used to refine the cDNA analysis?

7. The authors need to further expand on the important possibility that the loci derived from their experiments (having been derived from a laboratory cross), may not reflect the locus/loci underlying

CQR in natural populations of *P. vivax*. On this note, the authors have been somewhat biased in their reference to the literature on CQR in *P. vivax*. They fail to include an important reference to a study by Pava and colleagues (<https://www.ncbi.nlm.nih.gov/pubmed/26525783>), which found that *pvcr-t-o* expression was related to parasite stage but not *ex vivo* CQ sensitivity. The study by Pava and colleagues could be considered more robust than the study by Melo and colleagues (reference no. 13) since it uses robust *ex vivo* data on *P. vivax* isolates collected from a study site in Papua Indonesia with well documented evidence of CQR. The authors might also mention that a couple of population genomic studies conducted on *P. vivax* isolates in regions with good evidence of CQR, including Papua Indonesia (Pearson et al, ref 9) and Malaysia (Auburn et al - <https://www.ncbi.nlm.nih.gov/pubmed/29968722>), did not find evidence of signals of selection in the *pvcr-t-o* region.

8. Did the authors observe the CQR-associated microsatellite and *pvcr-t-o* intron 9 alleles in AMRU1 and/or other isolates (aside from the NIH-1993 strains) with evidence of CQR?

NCOMMS-19-05144-T

"*Plasmodium vivax* chloroquine resistance links to *pvcr*t transcription in a genetic cross" by Sá et al.
Responses to Reviewers' comments

Reviewer #1 (Remarks to the Author):

This manuscript by Sa et al. documents a quite remarkable investigation to identify a marker of
chloroquine (CQ) resistance in the human malaria parasite *Plasmodium vivax*. This has been a long
sought after goal and here the authors tackle this by implementing a genetic cross between *P. vivax*
parasites in a splenectomized chimpanzee. Because *P. vivax* blood stage parasites cannot be cultured
in vitro, the authors use CQ pressure and linkage group selection to scan the genome in the search for
a region that is positively selected in pools of CQ-selected progeny (passaged in Aotus or Saimiri
monkeys). Using a panel of 37 microsatellite markers, they identify enrichment of a chromosome 1
region containing the *P. vivax crt* (*pvcr*t) ortholog of *pfcr*t, the CQ resistance determinant in *P.*
*falciparum*. RT-PCR and protein expression data support increased expression of *pvcr*t in CQ-selected
progeny. Potential DNA changes in non-coding regions that might be causal for altered levels of gene
expression are identified. One complicating and significant twist in this study came with the finding
that the original CQ-resistant AMRU strain did not propagate in the chimpanzee, and that the two
parents were separate isolates existing in a mixed isolate NIH-1993, of which one was enriched by
sub-therapeutic CQ treatment.

*Response.* We thank the reviewer for these comments on the work.

The work behind this study is colossal and the authors deserve considerable credit for tackling this
very complex project. I believe that this article meets the level of interest and scientific originality and
robustness that merits publication in Nature Communications. There are however several
qualifications that I believe must be addressed in a revised manuscript.

**1.**One major comment is that the authors employ only 37 microsatellite markers across the *P. vivax*
genome, comprised of 14 chromosomes. There is therefore a reasonable chance that this low marker
density would not have detected other chromosomal regions that would have shown positive
selection in the CQ-treated progeny pools. This caveat should be clearly stated in the discussion and
the number of markers mentioned in the abstract.

*Response 1.1.* We preface our response by noting the distinction between mutations in two or more
separate loci that must be inherited together for the realization of a drug resistance phenotype vs. various
loci that can modulate the phenotype conferred by a central determinant of drug resistance. The referee's
comment presumably pertains to the former case of mutations in two or more required loci, and not to the
likelihood of mutations elsewhere that can modulate the level of resistance but are unessential to the
threshold phenotype of resistance.

For the case of two or more required loci, there is a chance that mutation of another locus was carried by
both parents or that only 37 microsatellite markers could have missed a region other than that of *pvcr*t on
chromosome #1. Linkage mapping of course wouldn't have been able to detect mutation of another locus
if the mutation was the same in both parents of the cross. Regarding the possibility that the 37 markers
could have missed a region, please refer to the map that was requested by Reviewer 2 and is provided
now as new Supplementary Figure 5. The map shows absences of microsatellite markers from large
regions of chromosome 8, 9, 12, 13, and 14. This situation is reminiscent of the early search for the
genetic basis of chloroquine resistance in *Plasmodium falciparum*, a search that involved 85 RFLPs on
that parasite's 14 chromosomes in the HB3×Dd2 cross [Wellems *et al.* 1991 *PNAS* 88: 3382-6]. In that

search, additional markers and demonstration of considerable linkage disequilibrium among individual
chromosome markers in the F1 progeny [average of approximately one crossover per chromosome per
meiosis; Jiang *et al.* 2011 *Genome Biol* 4: R33] strongly supported the conclusion that another required
determinant had not been missed, although modulating loci did influence the phenotype [Sa *et al.* *PNAS*
2009 106: 18883-9]. Similarly, the distribution of marker inheritance patterns on the *P. vivax*
chromosomes in NIH-1993-R×S suggests that an extraordinarily high level of recombination would have
been necessary in the vicinity of a postulated second required determinant for our markers to have missed
it. We agree that this possibility cannot be dismissed, but our experience with markers and recombination
in previous *Plasmodium* crosses (now including rodent as well as *P. falciparum* parasites) makes it pretty
unlikely that the chloroquine resistance threshold in NIH-1993-R×S required a second locus outside of
the chromosome region carrying *pvcrt*.

In the heat map of Figure 2a, marker MS15 on chromosome #5 shows the only other pattern with a strong
skew (8/10 = 80%) toward the absolute inheritance (10/10 = 100%) of the MS334 and In *pvcrt* markers
on chromosome #1. We asked if a second required chloroquine response determinant could be present in
the vicinity of MS15 on chromosome 5, but detailed analysis with SNPs across the entire chromosome
ruled out this possibility [Supplementary Figure 6]. Modifying loci that can alter the level of resistance
could well be present in NIH-1993-R×S progeny, just as they are for the HB3×Dd2 cross mentioned
above; however, a search for such loci would be far beyond the scope of the present work.

An important qualification of our study is that strains of *P. vivax* with higher levels of chloroquine
resistance may carry different determinants that are not present in the NIH-1993-R×S parasites.
Additional crosses and progress in laboratory methods, particularly *in vitro* culture, will be required to
search for such determinants.

*Manuscript revisions.*

A map of the 37 microsatellite markers is now provided in Supplementary Figure 5.

Text is added on pages 8–9 of the manuscript to address the possibility that an additional determinant
required for resistance might have been missed by the 37 markers and targeted genome sequencing:

“We note that our search would not have been able to map the presence of mutation that was the same
in both parents, nor does it dismiss the possibility that the 37 microsatellite markers could have missed
the inheritance of separate, additional determinant required for resistance. However, considering the
distribution of hotspots and recombination levels in previous *P. falciparum* crosses (average of
approximately one crossover per chromosome per meiosis²¹), an extraordinarily high level of
recombination activity would have been necessary in the vicinity of a postulated second determinant for
the markers to have missed it outside of the chromosome region carrying *pvcrt*.”

**2. Another major comment concerns their estimation of PvCRT expression levels based on one peptide**
**(IGNILEK). Were no other peptides from this protein identified or where those data too variable? This**
**needs to be clarified. I would have thought that predicting differences in protein expression levels**
**based on a single peptide is prone to inaccuracy.**

*Response 1.2.* In contrast to peptide identifications as they are assigned in shotgun proteomics, LC-
MS/MS identification followed by direct transition into LC-MRM quantification of *P. vivax* chloroquine
resistance transporter (PvCRT) involved processes that assure more rigorous accuracy. In these processes,
the selection of the single diagnostic peptide for PvCRT was necessary because of the limited sample
amounts and was based on criteria developed from our experience with a *P. falciparum* orthologous
peptide, unique to PfCRT that is not present in human red cell proteins or serum. Scaffold (Version 4.0.4,
Proteome Software) was used in an independent orthogonal analysis of a concatenated *P. vivax-Saimiri*
protein target decoy database to validate peptide choice, based on Mascot with the added strength of two
other MS search engines (X!Tandem and OMSSA). Scaffold implemented a decoy peptide search

function to ensure that the peptide matches were unique under stringent false discovery rates [1% False
Discovery Rate (FDR) for both the protein and peptide thresholds]. The results across the three search
engines confirmed that the IGNIILEK peptide is unique to PvCRT (99% protein probability and 100%
peptide probability) and is not shared with any other protein from *P. vivax* or from the *Saimiri* host red
cell or serum. The spectrum of this IGNIILEK peptide (acquired through an AB SCIEX TripleTOF™
5600 LC-MS/MS System) proved to be of high quality:

For specific PvCRT identification, the use of the isotope-labeled internal standard combined with the
rigorous choice of peptide assured us that only PvCRT was quantified from the limited samples having
low *P. vivax* densities. The heavy-atom labelled IGNIILEK peptide (6C13 on lysine) was spiked into each
*Saimiri* red cell lysate digestion product as an internal standard, and, in every experiment, this labeled
peptide showed the same chromatographic retention time as the native peptide derived from PvCRT.
Identical retention times provided another important criterion of experimental quality. Three transitions
providing maximal signals (low interference) were analyzed in each experiment demonstrating sequence
and assay specificity (Supplementary Table 13).

*Manuscript revision.* To address the reviewer's comment and improve the clarity of the manuscript, we
have incorporated the above points and added the following sentences to the Methods section:

"... we identified candidate tryptic peptides likely to be high quality, and which conformed with the
minimum general rules for the peptide's use in LC-MRM: no fewer than 8-amino acids; no known post-
translational modifications; no missed cleavages; and no predilection to artifactual modifications
(empirically determined by MS/MS analyses). Before use of the limited *P. vivax* samples, we performed
experiments with the orthologous PfCRT peptides from *P. falciparum* cultures, which provided larger
amounts of infected red cells and samples of greater scale. An orthologous peptide of PvCRT IGNIILEK
(the PfCRT peptide VGNILER; 85% sequence identity) met the quality thresholds for confident
identifications and was readily identified in replicate LC-MS/MS runs. Another PfCRT peptide
identified from our *in silico* predictions could also be observed, but it was less abundant, supporting
VGNILER as the preferred choice for PfCRT quantification in samples with low parasite densities. By
procedures described below, testing of samples from *P. vivax*-infected *Saimiri* confirmed that the
homologous PvCRT peptide, IGNIILEK, was readily detected at low parasitemias and provided the
requisite high quality spectrum for PvCRT quantification."

**3. Also, given that the authors could enrich for the CQ-resistant subpopulation with CQ treatment,**
**did they obtain a pure resistant population? Was it possible to generate whole-genome sequence**
**data from that parent? If so, those data should be deposited. Is the genome sequence data from the**
**sensitive parent too heterogeneous (because of the resistant subpopulation) to report?**

*Response 1.3.* The reviewer correctly surmises that sequence data from the sensitive parent were
heterogeneous, consistent with composition of multiple clones. As we discuss further in *Response 2.1*
below, this finding is consistent with other evidence that *P. vivax* infections commonly involve mixed
subpopulations of different parasite clones [Ferreira *et al.* 2007 *J Infect Dis*; Popovici *et al.* 2018 *MBio*].
Genome sequencing data from the NIH1993-S parent are now deposited in the NCBI Short Read Archive
under accession number SRR2148599.

In additional experiments, we attempted to obtain whole-genome sequence data from a pure resistant
parent population after chloroquine pressure. Unfortunately, only small amounts of parasite material
could be recovered, and the amplification and sequencing results did not provide interpretable, clean data
covering the genome. As the referee notes, heterogeneity of the mixed populations may have contributed
to the problems of interpretability. Additional material from monkey infections is not available to conduct
the sequencing again (monkeys are presently difficult to obtain).

*Manuscript revisions.*

“Genome sequencing data from the NIH1993-S parent are deposited in the NCBI Short Read Archive
under accession number SRR2148599.” is added to the *Data availability* statement.

Above points are also treated in a new paragraph detailed below in Response 2.1.

**Minor comments:**

**Line 140. The authors should clarify in the text that the MS334 and *pvcrt* intron 9 markers show**
**positive selection in all LGA pairs in the post-CQ samples.**

*Response.* Yes - the MS334 and *pvcrt* intron 9 markers indeed showed positive selection in all LGA pairs
in the post-CQ samples.

*Manuscript revision.*

“CQ selection *E* values displayed by heat map (**Fig. 2a, Supplementary Table 9**), showed increases (*E*
> 1) as well as decreases (*E* < 1) among the LGS pairs for every marker except the two specific to
chromosome 1: MS334 and a *pvcrt* intron 9 polymorphism (In9*pvcrt*), which showed positive selection
in all LGS pairs in the post-CQ samples.”

**Line 192. Should be “breakthroughs in culture”**

*Response/manuscript revision.* Corrected to “...breakthroughs in culture and genetic manipulation.”

**Line 229 is awkwardly worded: “parental lines blood-stage”.**

*Response/manuscript revision.* Rewritten “...infection inoculated with blood-stages from the parental
lines...”

**Figure 2b needs a scale on the X axis.**

*Response/manuscript revision.* The requested scale is added to Figure 2b. A scale is also added to the
panel of Supplementary Figure 6.

**Figure 3 focuses on TGAAGH repeats, as does the text. Evidence that this might be significant**
**comes from studies in other systems. Is there any evidence that this element might be important in**

**any Plasmodium species? If not then that should be stated. It is clear that such an experiment**
**cannot be performed in P. vivax at present.**

*Response.* In answer to the reviewer’s question, there is no evidence that TGAAGH repeats might have
an importance in *Plasmodium* species analogous to their importance in the other systems. The revised
manuscript now states this as suggested. We also note that transcripts containing intron 9 have been
recently reported with a predicted early stop codon after the repeats [Kim *et al.* 2017 *Sci Rep*];
interestingly, no analogous introns with TGAAGH repeats occur in the *crt* genes of other major
*Plasmodia* clades, including those of *P. falciparum* and *P. knowlesi*.

*Manuscript revision.*

“Interestingly, no analogous introns with TGAAGH motifs occur in the *crt* genes of other major
*Plasmodia* clades, including those of *P. falciparum* and *P. knowlesi*; and *P. vivax* transcripts containing
intron 9 sequences have been reported with a predicted early stop codon after the repeats²⁶. Although
there is currently no knowledge regarding their role in *Plasmodium*, TGAAGH motifs are involved in
gene regulation by *Arabidopsis*, including the activity of cis-acting regulatory elements or CAREs²⁷,
intron-mediated enhancers²⁸, and splicing enhancer motifs ...”

**Figure 3a also needs numbers of independent and technical repeats listed.**

*Response.* Figure 3a is based on data collected from five monkeys, each infected with a sample
(subpopulation) from the progeny pool. Since each monkey was unique, and since the subpopulations
contained different mixtures of unique as well as common progeny, no two infections were identical, and
each drug selection experiment was considered independent. Calculations from PCR determinations were
performed on results collected from two technical replicates.

*Manuscript revision.* The above information is now incorporated into the legend of Figure 3a: “Error bars
indicate standard deviations from two technical replicates.”

**Line 490. I think this should read 10 chromosomes, not 14.**

*Response/manuscript revision.* Text corrected, thank you: “... 37 polymorphic microsatellite markers
measured on 14 chromosomes.”

**Figure S2: I would indicate that the Malarone treatment was over multiple days.**

*Response/manuscript revision.* Legend revised as suggested: “Malarone treatment was administered for
three days (arrow indicates day of first dose) to the chimpanzee after mosquito feeding 6 to cure the
infection. Parasitemia values are provided in Supplementary Table 3.”

**Reviewer #2 (Remarks to the Author):**

**The manuscript by Sa and colleagues attempts to address an important knowledge gap concerning**
**Plasmodium vivax malaria – specifically, determining the molecular determinant(s) of chloroquine**
**resistance (CQR) in this species. An undoubtedly huge amount of work has gone into the study and**
**the authors have made a concerted effort to pull together experts from various disciplines to aid in**
**different aspects of the experimental, laboratory and data analysis components of the study. There**
**are unfortunately still some limitations that somewhat constrain the confidence of the results. Further**
**discussion/acknowledgment of these limitations (described further below) is needed to give readers**
**more clarity in interpreting the data.**

**1. The most substantial limitation is the initial inability to generate progeny between the intended**
**parental lines, AMRU1(CQR) and NIH-1193 (CQS). As a result, the project appears to be constrained**
**from the start to investigating the progeny of a more accidental cross using the CQS NIH-1993 strain**
**(NIH-1993-S) and a derived strain, which the authors claim is CQR (NIH-1993-R). Unless I'm mistaken,**
**it appears that the initial basis for defining NIH-1993-R as CQR (and for going ahead with the cross)**
**appears to be that this strain was not detected until after the host was subject to a sub-therapeutic**
**dose of CQ. However, it should be acknowledged that this is only an assumption. The possibility**
**remains that the NIH-1993-R strain is CQS and simply took longer to amplify owing to a much lower**
**initial parasite density and/or other growth-related phenotypic variation relative to NIH-1993-S.**
**Indeed, the CQS sensitive NIH-1993-S strain appears to have been able to persist through the**
**sub-therapeutic CQ dose fairly well. It would be helpful for the authors to discuss the afore-mentioned**
**limitations further to give the reader more clarity on the potential issues.**

*Response 2.1.* Recent studies strengthened the evidence that *P. vivax* infections commonly involve mixed
subpopulations of different parasite clones [Ferreira *et al.* 2007 *J Infect Dis*; Popovici *et al.* 2018 *MBio*].
Mixed clonal subpopulations in the *P. vivax* Sal-I is thus not an unusual finding. This fact accounts for
observations that the parental NIH-1993-S and NIH-1993-R lines have closely related genotypes and yet
are distinct from each other and from some earlier Sal-I populations by markers including CSP types,
*pvdhfr-ts* mutations, and microsatellites of *pvcrt*. Passages through different monkey systems, and other
pressures such as drug exposure, would have caused the prevalence of various parasite subpopulations to
rise and fall relative to one another over time.

The genetic relationships among the subpopulations of Sal-I reflect the recombinant genotypes in
sporozoites transmitted by mosquitos. Meiosis events that produced these sporozoites thus dictated the
inheritance of genetic material from ancestral infections as well as the genotypes and relationships of the
NIH-1993-S and NIH-1993-R lines. Meiosis giving rise to these sporozoites also may account for the
paucity of distinguishable microsatellites markers across the large regions of some NIH-1993-S and NIH-
1993-R chromosomes, despite our search of 232 pre-qualified candidate repeats for such markers.

It is perhaps not surprising that our study found pre-existing differences among the Sal-I and NIH-1993
subpopulations in two genes involved in drug response: *pvdhfr-ts* and *pvcrt*. These differences include
codon mutations in *pvdhfr-ts* associated with pyrimethamine resistance, and microsatellite polymorphisms
in *pvcrt* associated with gene expression and chloroquine treatment failure. The evidence for these
differences in subpopulations of Sal-I (*ergo* certain NIH-1993 subpopulations) likely reflects historical
exposures of ancestral parasites to these antimalarial drugs. Intriguingly, without chloroquine pressure,
the NIH-1993-S type was strongly favored over NIH-1993-R in the monkey infections of our study. We
speculate that the genetic changes *P. vivax* parasites need to survive chloroquine might reduce their ability
to complete the life cycle when there is no drug pressure, a possibility of reduced fitness that may help to
explain the slow rise and spread of chloroquine-resistant *P. vivax* relative to *P. falciparum* in malaria-
endemic regions.

With the above considerations in mind, and in view of the experimental evidence that upregulated
expression of PvCRT confers chloroquine resistance in *P. vivax* as well as heterologous systems, we
respectfully disagree with the reviewer's suggestion that NIH-1993-R strain is CQS and simply took
longer to amplify owing to a much lower initial parasite density and/or other growth-related phenotypic
variation relative to NIH-1993-S. However, we agree that the CQS sensitive NIH-1993-S strain was able
to persist through the sub-therapeutic chloroquine dose fairly well, and that other *P. vivax* strains could
have higher levels of chloroquine resistance than NIH-1993-R by different mutations than those present in
the cross.

*Manuscript revision.* A new paragraph in the manuscript addresses the above points and also the
questions raised in Review Comment 1.3.

“Our results leave open possibilities that determinants in addition to *pvcr*t may be involved in *P. vivax*
CQR or that strains of *P. vivax* with higher levels of resistance may carry mutations not present in the
NIH-1993-R×S parasites. The effect of *pvcr*t upregulation is consistent nonetheless with greater
expression levels in patients with CQ-resistant vivax malaria^{13,30}. To the extent that greater expression
may reverse in the absence of drug pressure or not maintain well under laboratory conditions, *pvcr*t
upregulation may be difficult to detect in studies of different design, for example in population surveys
or *ex vivo* investigations of resistance^{9,31,32}. A fortuitous aspect of our present work was the crossing of
two distinct subpopulations from the NIH-1993 line, of which one was enriched by sub-therapeutic CQ
treatment in the chimpanzee. *P. vivax* infections are commonly multiclonal^{33,34}, helping to explain why
NIH-1993-S and NIH-1993-R have related genotypes, yet are distinct from each other and from other
Sal-I populations by markers including CSP types, *pvdhfr-ts* mutations, and microsatellites of *pvcr*t.
Passages of Sal-I and NIH-1993 through different host systems, and pressures such as drug exposure,
would have caused the prevalence of various parasite subpopulations to change over time. Intriguingly,
without chloroquine pressure, NIH-1993-R type parasites seem to have been less favored than NIH-
1993-S in the monkey infections of our study. We speculate that the genetic changes *P. vivax* parasites
need to survive chloroquine reduce their fitness relative to CQ-S parasites in the absence of drug
pressure, a hypothesis that might help to explain the slow rise and spread of CQ-R *P. vivax* relative to
*P. falciparum* in malaria-endemic regions.”

**2. Another limitation of the study is the limited (and somewhat biased) representation of the genome**
**by the microsatellite marker set used in the linkage group selection analysis. While some**
**chromosomes have several markers, others (chrs 2,4,7,9 and 11) are only represented by a single**
**marker – this would surely constrain the ability to detect signals on those chromosomes. How were**
**the 37 microsatellites selected? The authors might also consider adding a figure showing the genomic**
**locations of the markers.**

*Response 2.2.* These questions of microsatellite marker representation and the detection of signals are
also raised in first comment of Reviewer 1. In Response 1.1, we suggest from previous work that an
extraordinarily high level of chromosomal recombination would have been necessary for the 37
microsatellite markers to have missed the required inheritance of separate locus with a resistance mutation
differing between the two parents. We agree that this possibility cannot be entirely dismissed, but our
experience with markers and recombination in previous *Plasmodium* crosses (now including rodent as
well as human parasites) suggests little likelihood that the chloroquine resistance threshold in NIH-1993-
R×S required a second locus outside of the region of chromosome #1 carrying *pvcr*t.

We appreciate the suggestion to include a map showing the genomic locations of the 37 markers and have
added this in a Supplemental Figure. Because of the close genetic relationship of the NIH-1993-S and
NIH-1993-R populations (response 2.1 above), it was necessary for us to search 232 genome-wide
microsatellite sequences to identify the 37 markers that could distinguish chromosome inheritance in the
NIH-1993-S×R cross. In earlier studies, those microsatellite sequences had distinguished eight *P. vivax*
laboratory lines [Carlton *et al.* 2008; Feng *et al.* 2003] or had been used to discriminate among *P. vivax*
parasites in population studies [Karunaweera *et al.*, 2007; Joy *et al.*, 2008].

*Manuscript revisions.* A map showing the locations of the microsatellite markers is added as
Supplemental Figure 5. The four references for the 232 microsatellite sequences are provided in the
Methods Section. Additional manuscript revisions pertaining to microsatellite marker representation and
the detection of signals are detailed in Responses 1.1 and 2.1.

3. Following on from comment 2, it is a shame that the authors restricted the genomic analysis to chromosomes 1 and 5. Is this because there wasn't sufficient DNA even though they used WGA? For clarity for the readers, the authors might consider openly noting this is a limitation requiring further exploration in future studies.

Response 2.3. Although attempts were made to perform a genome-wide scan for resistance, sequence efforts failed on the remaining DNA after chromosome 1 and 5 specific sequencing. The Reviewer rightly surmises that insufficient DNA remained for additional attempts despite the use of WGA. However, please note Response 1.1 about the low likelihood that the chloroquine resistance threshold in NIH-1993-R×S required a second locus outside of the chromosome 1 region carrying *pvcr1*.

Manuscript revision. Text addressing the above points added to the manuscript:

“On chromosome 5, which contains a microsatellite marker (MS15) with *E* values > 1 in 8/10 recrudescence infections, targeted genome sequencing detected no region significantly selected by CQ (**Supplementary Fig. 6**; insufficient DNA was available to complete similar sequencing of the other chromosomes). We note that our search would not have been able to map the presence of mutation that was the same in both parents, nor does it dismiss the possibility that the 37 microsatellite markers could have missed the inheritance of separate, additional determinant required for resistance. However, considering the distribution of hotspots and recombination levels in previous *P. falciparum* crosses (average of approximately one crossover per chromosome per meiosis²¹), an extraordinarily high level of recombination activity would have been necessary in the vicinity of a postulated second determinant for the markers to have missed it outside of the chromosome region carrying *pvcr1*.”

4. Why were different hosts (some *Aotus* and some *Saimiri*) used in the drug response analysis?

Response 2.4. Although *Aotus* and *Saimiri* are both well-known for their ability to serve as nonhuman primate hosts for *P. vivax*, we were uncertain if the host species would affect the treatment recrudescences differently. Our finding of recrudescences in both model systems allowed us to analyze a larger number of experimental infections in face of limited availability of monkeys (particularly *Aotus*).

5. Why did the total dose of CQ used in the drug response analysis vary (from 15-55 mg/kg total)?

Response 2.5. Once it was clear that the NIH-1993-R phenotype could recrudescence after 15 mg/kg total dose (adequate plasma levels verified), we pushed the system to as much as 55 mg/kg total to test for upper boundaries of the phenotype.

6. I appreciate the authors' note on line 371 that cryopreserved samples were thawed by methods that should in theory favour ring stages, however, was this confirmed with microscopy data? If so, was the data on the percentage of rings used to refine the cDNA analysis?

Response 2.6. A few studies have demonstrated favored survival of ring over trophozoite stages after cryopreservation. Margos *et al.* (<https://www.ncbi.nlm.nih.gov/pubmed/1598502>) used synchronized cultures of *P. falciparum* to demonstrate much better survival of ring stages (young trophozoites) relative to later stage trophozoites and schizonts, which could only be somewhat protected by a freezing program designed for lymphocytes. Collection of survival data required return of the thawed parasites to culture for follow-up over ensuing growth cycles, a strategy not possible with *P. vivax* parasites because they cannot be maintained in culture. Although we observed good morphological appearances of *P. vivax* ring stages among few intact and healthy appearing later stages in scans our thawed samples, the material was

380 very limited (monthly maximum total draw of 3 mL/kg per animal) and we could not use it at the expense
of other preparations for more extensive morphological examinations and counting.

**7. The authors need to further expand on the important possibility that the loci derived from their**
**experiments (having been derived from a laboratory cross), may not reflect the locus/loci underlying**
**CQR in natural populations of *P. vivax*. On this note, the authors have been somewhat biased in their**
**reference to the literature on CQR in *P. vivax*. They fail to include an important reference to a study by**
**a Pava and colleagues (<https://www.ncbi.nlm.nih.gov/pubmed/26525783>), which found that *pvcr-t-o***
**expression was related to parasite stage but not *ex vivo* CQ sensitivity. The study by Pava and**
**colleagues could be considered more robust than the study by Melo and colleagues (reference no. 13)**
**since it uses robust *ex vivo* data on *P. vivax* isolates collected from a study site in Papua Indonesia**
**with well documented evidence of CQR. The authors might also mention that a couple of population**
**genomic studies conducted on *P. vivax* isolates in regions with good evidence of CQR, including**
**Papua Indonesia (Pearson et al, ref 9) and Malaysia (Auburn et al -**
**<https://www.ncbi.nlm.nih.gov/pubmed/29968722>), did not find evidence of signals of selection in the**
***pvcr-t-o* region.**

*Response 2.7.* Pava et al. [2016] did not find any correlation between *pvcr-t* (*pvcr-t-o*) transcription levels
and CQ resistance in *ex vivo* assays, however, other work in addition to Melo et al. [2014] reported
correlations of higher *pvcr-t* transcription to CQ resistance in patients [Fernandez-Becerra, 2009]. These
reports complemented the earlier evidence for *pvcr-t* expression and phenotypes of reduced chloroquine
response in heterologous models of amoeba and *P. falciparum* [Sa et al, 2006]. Could the differing results
from the *ex vivo* study of Pava et al. vs. the *in vivo* findings from non-human primate and patient
infections (as well as the phenotypes of *pvcr-t* in heterologous systems) relate to the untoward effects of *ex*
*vivo* culture conditions on *P. vivax* drug response? We suspect so, but a definitive answer to this question
remains undeveloped in the absence of *in vitro* culture conditions for the propagation of *P. vivax*.

Thank you for the suggestion to add that no *pvcr-t* selection signals were found in the population studies of
Pearson et al. [2016] and Auburn et al. [2018].

*Manuscript revision.* Sentences have been added to the revised manuscript to treat the above points and
include the recommended references:

“Our results leave open possibilities that determinants in addition to *pvcr-t* may be involved in *P. vivax*
CQR or that strains of *P. vivax* with higher levels of resistance may carry mutations not present in the
NIH-1993-R×S parasites. The effect of *pvcr-t* upregulation is consistent nonetheless with greater
expression levels in patients with CQ-resistant vivax malaria^{13,30}. To the extent that greater expression
may reverse in the absence of drug pressure or not maintain well under laboratory conditions, *pvcr-t*
upregulation may be difficult to detect in studies of different design, for example in population surveys
or *ex vivo* investigations of resistance^{9,31,32}.”

**8. Did the authors observe the CQR-associated microsatellite and *pvcr-t-o* intron 9 alleles in AMRU1**
**and/or other isolates (aside from the NIH-1993 strains) with evidence of CQR?**

*Response 2.8.* We have collected preliminary data from AMRU-I and find that the 5’ flanking region and
intron of *pvcr-t* contain the repeats with a complexity of many differences from the NIH-1993 patterns.
Mapping of these repeats and differences to CQ responses, and exploration of mechanisms in untranslated
regions of *pvcr-t* that interact with transcription factors, will require methods of investigation unavailable
at the present time, particularly methods of continuous culture *in vitro*.

Reviewers' Comments:

Reviewer #1:

Remarks to the Author:

The corresponding author has responded well to the comments from the first round of review through textual changes that improve the manuscript. There are just two instances where I feel the response should be amended:

1. In Response 1.1 the author refers to earlier *P. falciparum* crosses when stating "However, considering the distribution of hotspots and recombination levels in previous *P. falciparum* crosses (average of approximately one crossover per chromosome per meiosis), an extraordinarily high level of recombination activity would have been necessary in the vicinity of a postulated second determinant for the markers to have missed it outside of the chromosome region carrying *pvcrt*." Can the author provide more accuracy from earlier crosses in terms of a more accurate mean \pm SEM. He should also consider the published cross data from Vaughan et al (CIDR, Seattle). It may be that the average is much closer to 2. More importantly, the author should still explicitly acknowledge the low marker coverage on some chromosomes, notably 9 and 13 (see Figure S5) where there is a sole marker each and therefore a reasonable chance that a locus could have been missed. The author should also replace "extraordinarily" with for example "particularly". Of note, the frequency of meiotic crossover events has not been studied in detail in *P. vivax*.

2. In Response 2.7 the author should more explicitly state that Pava et al 2016, Pearson et al 2016 and Auburn et al 2018 did not find evidence of an association between the *pvcrt-o* region and chloroquine resistance in *P. vivax*. As it stands the revised manuscript cites these in an indirect way: "To the extent that greater expression may reverse in the absence of drug pressure or not maintain well under laboratory conditions, *pvcrt* upregulation may be difficult to detect in studies of different design, for example in population surveys or ex vivo resistance [9,31,32]."

Reviewer #2:

Remarks to the Author:

Thank you for providing detailed answers to all of my queries. I have no further questions or comments.

NCOMMS-19-05144-B “*Plasmodium vivax* chloroquine resistance links to *pvcr1* transcription in a genetic cross” by Sá et al.

Responses to Reviewers' comments:

REVIEWERS' COMMENTS:

Reviewer #1 (Remarks to the Author):

The corresponding author has responded well to the comments from the first round of review through textual changes that improve the manuscript. There are just two instances where I feel the response should be amended:

1. In Response 1.1 the author refers to earlier *P. falciparum* crosses when stating “However, considering the distribution of hotspots and recombination levels in previous *P. falciparum* crosses (average of approximately one crossover per chromosome per meiosis), an extraordinarily high level of recombination activity would have been necessary in the vicinity of a postulated second determinant for the markers to have missed it outside of the chromosome region carrying *pvcr1*.” Can the author provide more accuracy from earlier crosses in terms of a more accurate mean \pm SEM. He should also consider the published cross data from Vaughan et al (CIDR, Seattle). It may be that the average is much closer to 2. More importantly, the author should still explicitly acknowledge the low marker coverage on some chromosomes, notably 9 and 13 (see Figure S5) where there is a sole marker each and therefore a reasonable chance that a locus could have been missed. The author should also replace “extraordinarily” with for example “particularly”. Of note, the frequency of meiotic crossover events has not been studied in detail in *P. vivax*.

We thank the reviewer for these additional comments.

Response.

In the absence of information about the frequency of meiotic crossover events in *Plasmodium vivax*, data is available from two other *Plasmodium* species: *Plasmodium falciparum* and *Plasmodium chabaudi chabaudi*, a rodent malaria parasite evolutionarily closer to *P. vivax* than is *P. falciparum* (Loy et al. 2017 *Int J Parasitol*). Exclusive of highly recombinogenic subtelomeric (accessory) regions, genome genetic lengths are reported to be 1655 cM and 1570 cM for *P. falciparum*, and 1676 cM for *P.c. chabaudi* (Jiang et al. 2011 *Genome Biol*; Miles et al. 2016 *Genome Res*; Martinelli et al. 2005 *Malaria J*). An estimate of 2210 cM by Vaughan et al. (*Nat Methods* 2015) did not distinguish the contributions of markers from hypervariable regions including gene families involved in processes of antigenic variation. For the purposes of our present discussion, the core genetic lengths are appropriate, indicating 16.55, 15.70, and 16.76 crossovers per generation, by definition of the cM. These yield averages of 1.19, 1.12, and 1.20 crossover per chromosome, although individual numbers vary for each of the 14 chromosomes because of their different physical lengths (Fig. 5 of Martinelli et al.; Fig. 3 of Miles et al.). Considering this fact and that chromosome organizations differ among the *Plasmodium* species, we are hesitant to calculate and try to interpret a mean \pm SEM; instead, the above summary and references are added to the manuscript for the reader's information and interpretation.

We appreciate and agree with the recommendation to replace “extraordinarily” with “particularly”.

Manuscript revisions. Text on pp 8–9 is expanded to address the reviewer's comments and include the information above:

“Although the frequency of meiotic crossover events is not reported for *P. vivax*, information is available from two other *Plasmodium* species: *P. falciparum* and *Plasmodium chabaudi chabaudi*, a rodent malaria parasite that is evolutionarily closer to *P. vivax* than to *P. falciparum*²². Excluding the highly recombinogenic subtelomeric regions, genome genetic lengths of 1655 and 1570 centiMorgans (cM) have been reported for *P. falciparum*^{23,24}, and 1676 cM for *P.c. chabaudi*²⁵. Another estimate of 2210 cM for *P. falciparum* did not distinguish the contributions of markers from subtelomeric regions including gene families involved in processes of antigenic variation²⁶. For the purposes of our present discussion, the core genetic lengths are appropriate, indicating 16.55, 15.70, and 16.76 crossovers per generation by definition of the cM. These yield averages of 1.19, 1.12, and 1.20 crossover per chromosome, but individual numbers vary with chromosome size^{23,24,25}. We suggest that a particularly high level of recombination activity would have been necessary in the vicinity of a postulated second determinant for the markers to have missed it outside of the chromosome region carrying *pvcr*.”

2. In Response 2.7 the author should more explicitly state that Pava et al 2016, Pearson et al 2016 and Auburn et al 2018 did not find evidence of an association between the pvcr-o region and chloroquine resistance in P. vivax. As it stands the revised manuscript cites these in an indirect way: “To the extent that greater expression may reverse in the absence of drug pressure or not maintain well under laboratory conditions, pvcr upregulation may be difficult to detect in studies of different design, for example in population surveys or ex vivo resistance [9,31,32].”

Response. Other studies have identified changes in the *pvcr* 5'UTR and introns associated with selective evolutionary pressure on putative drug resistance genes (Flannery *et al.* 2015 *ACS Infect Dis*; Dharia *et al.* 2010 *PNAS*). Detection of *pvcr* changes in these surveys stands in distinction to the findings of Pava *et al.*, Pearson *et al.*, and Auburn *et al.*

Manuscript revisions. Our manuscript is revised to take the reports of Flanner *et al.* and Dharia *et al.* into account along with the comments of the referee:

Introduction Section, pp 3–4: “Increased transcript levels of *pvcr* and the *P. vivax* multidrug resistance gene (*pvmdr-1*) have been reported in samples from patients with severe *P. vivax* malaria¹³ or who suffered CQ treatment failures¹⁴. In contrast, another study found that *pvcr* transcription was related to parasite stage but not to *ex vivo* CQ susceptibility in patient samples¹⁵.”

Discussion Section, pg 11: “These results may help to explain evidence of selective pressure in the 5'UTR and intronic regions of *pvcr* in previous studies^{35,36,37}, including occurrences in the top 1% of genes with intronic mutations, although others found no association between the *pvcr* genomic region and *P. vivax* CQR^{9,38}. Our evidence for upregulated *pvcr* transcription is also consistent with some findings from patients with CQ-resistant vivax malaria^{13,14}. However, such upregulation was not observed in *ex vivo* assays of patient samples¹⁵. To the extent that greater expression may reverse in the absence of drug pressure or not maintain well under laboratory conditions, *pvcr* upregulation may be difficult to detect in *ex vivo* studies.”

Reviewer #2 (Remarks to the Author):

Thank you for providing detailed answers to all of my queries. I have no further questions or comments.

Response. We thank the reviewer for the effort and comments that helped to improve our report.